# Discontinuity-Preserving Image Super-Resolution via MAP-Regularized One-Step Diffusion

## Abstract

We propose a real-world image super-resolution framework that leverages a pre-trained text-to-image Stable Diffusion model optimized for single-step sampling. Unlike traditional multi-step diffusion-based methods, which are computationally intensive, our approach enables fast inference while preserving high perceptual quality. To this end, we integrate a lightweight image enhancement module trained jointly with the diffusion model under a Maximum A Posteriori (MAP) formulation. The optimization includes a compound Markov Random Field (MRF) prior, derived from the anticipated discontinuity line field energy, which functions as a structural regularizer to preserve fine image details and facilitate deblurring. Existing single-step diffusion approaches often rely on distillation or noise map estimation, which limits their ability to generate rich pixel-space details. In contrast, our method explicitly models high-frequency line field consistency between the low- and high-resolution domains, guiding the image enhancer to reconstruct sharp outputs. By preserving and enhancing structural features such as edges and textures, our framework effectively handles complex degradations commonly encountered in real-world scenarios. Experimental results demonstrate that our method achieves performance that is comparable to or exceeds that of state-of-the-art single-step and multi-step diffusion-based image super-resolution methods qualitatively, quantitatively, and computationally.

## 1 Introduction

Image super-resolution (ISR) Chen et al. (2023a); Liang et al. (2022a); Wang et al. (2018); Liang et al. (2021); Zhang et al. (2018; 2022); Geman & Geman (1984) is a fundamentally important and inherently ill-posed inverse problem that has been actively studied since before the advent of deep learning, and continues to remain a compelling and challenging research topic. The objective of ISR is to reconstruct a high-quality (HQ) image from its corresponding low-quality (LQ) counterpart, which typically suffers degradation due to factors such as noise, blur, and aliasing—leading to the loss of high-frequency details critical for perceptual quality. Early ISR methods Dong et al. (2014); Liang et al. (2021); Zhang et al. (2022; 2018) typically assumed a simplified degradation model comprising a fixed sequence of operations—namely, blurring, downsampling, and the addition of white Gaussian noise. However, such models often fail when confronted with complex, unknown degradations, particularly when the degradation process is nonlinear or contains highly uncertain combinations of distortions. To address this limitation and move closer to real-world scenarios, the field has increasingly shifted toward real-world image super-resolution (Real-ISR) Zhang et al. (2021); Wang et al. (2021), which acknowledges that real image degradations are far more complex and diverse. In this setting, reconstructing an HQ image from an LQ input becomes significantly more challenging due to substantial information loss—especially in the high-frequency content of RGB images and aliasing. To simulate such real-world degradations during training, pioneering works such as BSRGAN Zhang et al. (2021) and Real-ESRGAN Wang et al. (2021) proposed degradation pipelines involving multiple sequential high-order distortions applied in random order. As a result, the trained model can better generalize to unseen, complex degradations and infer a plausible high-resolution reconstruction. The core objective of real-ISR in such settings is to remove blur, suppress noise, and perform accurate upsampling with anti-aliasing, thereby recovering fine

structural details. It is now well recognized that training with simple pixel-wise loss functions is insufficient; although such losses may reduce noise and perform upsampling, they often result in overly smoothed and perceptually unconvincing reconstructions Ledig et al. (2017); Wang et al. (2018). To effectively capture the statistics of natural HQ images, early deep learning-based ISR methods introduced various architectural innovations Dong et al. (2018) along with specialized loss functions. With the rise of generative models, particularly Generative Adversarial Networks (GANs), the super-resolution community adopted adversarial training frameworks for real-ISR Ledig et al. (2017); Wang et al. (2021); Liang et al. (2022a); Wei et al. (2020). In this paradigm, the generator network is trained to produce HQ images, while the discriminator evaluates the realism of these outputs, encouraging the generator to synthesize images that closely approximate natural textures and structures. The incorporation of GANs brought substantial improvements in visual fidelity and realism of super-resolved outputs. However, adversarial training also introduced new challenges: the generated images sometimes contained hallucinated details or artifacts that were inconsistent with the underlying ground truth, potentially deviating from the true image content and introducing misleading features.

The subsequent evolution of generative models has been significantly shaped by diffusion models Ho et al. (2020); Song et al. (2020b), which have gained prominence due to their superior training stability and more reliable image generation compared to GANs. The adaptation of diffusion models to operate in latent space has further enabled high-resolution image synthesis, while introducing modular conditioning mechanisms—such as text, sketches, or semantic maps—that enhance control over the generative process Rombach et al. (2022); Saharia et al. (2022). As a result, super-resolution tasks have also benefited from these advancements, enabling finer control over image restoration with an emphasis on preserving specific features. Among diffusion-based frameworks, Stable Diffusion (SD) Rombach et al. (2022) stands out for being trained on a large-scale dataset of text-image pairs, thereby capturing rich natural image priors. Its capacity to generate photorealistic images has opened up opportunities to adapt and modularize the model for Real-ISR. Building on this, several methods Wang et al. (2024a); Lin et al. (2024); Wu et al. (2024b); Yang et al. (2024); Yu et al. (2024) have leveraged pretrained SD pipelines to improve the perceptual realism and structural fidelity of Real-ISR outputs, pushing the performance beyond the limitations of GAN-based approaches. Despite these advancements, a significant limitation of diffusion-based methods lies in the slow inference time and uncontrolled image sharpening inherent to DDPM Ho et al. (2020). Achieving high-fidelity image generation typically necessitates a large number of iterative denoising steps, resulting in a computationally intensive and time-consuming sampling process during inference—an undesirable characteristic for practical applications. Although alternatives like the DDIM Song et al. (2020a) mitigate this by removing the Markovian assumption and reducing sampling time, they often compromise image quality, especially when using fewer sampling steps. Thus, there exists an inherent trade-off between sampling efficiency and the perceptual quality of the generated images in diffusion-based Real-ISR.

To strike a balance between inference speed and output image quality, we propose **D**iscontinuity Preserving **MAP**-optimized Image **S**uper-**R**esolution (**DMAPSR**), a framework that enables high-quality image generation using a single diffusion step. This is achieved by introducing an additional module, termed the image quality enhancer, which operates alongside a pretrained SD model. To address the challenge of oversmoothed or low-detail outputs typically associated with fast inference in diffusion models, we incorporate a Markov Random Field (MRF) energy term with appropriate relaxation to the original image content into the pipeline. This MRF prior acts as a structural regularizer, encouraging the preservation of fine-grained details in the low-quality input and promoting alignment with known natural image priors learned by the pretrained model. Specifically, the line-field based regularization within the MRF prior enforces the retention of important discontinuities and edges, ensuring that critical structures in the low-resolution image are preserved and enhanced in the final output. The image quality enhancer is trained jointly with the frozen noise predictor from SD, optimizing a loss that encourages the corrected sample to yield accurate noise estimates and visually rich reconstructions. This combination enables a single-step sampling process that significantly accelerates inference while maintaining fidelity and perceptual quality. Extensive experiments demonstrate that our approach achieves strong quantitative and qualitative performance in super-resolution tasks, still offering over a $100\times$ speedup compared to conventional multi-step diffusion models.

## 2 RELATED WORK

**Deep learning and GAN-based ISR.** Early deep learning-based ISR methods Chen et al. (2021; 2023b;a); Dai et al. (2019); Zhang et al. (2022) primarily addressed the problem under fixed and simplistic degradation models, which limited their applicability in real-world scenarios. In response to the need for more realistic modeling, BSRGAN Zhang et al. (2021) and Real-ESRGAN Wang et al. (2021) introduced more sophisticated GAN-based frameworks designed to handle complex and diverse degradation patterns encountered in practice. These advancements led to a notable improvement in visual quality and subsequently inspired a series of follow-up studies Chen et al. (2022a); Liang et al. (2022a;b); Xie et al. (2023) exploring variations of GANs tailored for real-ISR. While these approaches demonstrated improvements, GAN-based models inherently suffer from instability during training due to the adversarial learning framework, which involves simultaneous optimization of generator and discriminator networks. Additionally, the HQ images produced often contain unnatural textures and hallucinated artifacts, which undermine their fidelity and realism.

**Diffusion prior for real-ISR.** Diffusion models, formulated either through stochastic differential equations (SDEs) Song et al. (2020b) or denoising diffusion probabilistic models (DDPMs) Ho et al. (2020), have demonstrated impressive results in text-to-image generation and have subsequently been adapted for a variety of image restoration tasks. With the emergence of SD, which leverages latent-space modeling and text-conditioned priors, the pretrained text-to-image (T2I) Stable Diffusion pipeline Rombach et al. (2022) has been increasingly adopted for Real-ISR tasks Lin et al. (2024); Wang et al. (2024a); Yang et al. (2024); Wu et al. (2024b); Yu et al. (2024). Some of these methods generate high-resolution images directly from noise using fine-tuned adapters Zhang et al. (2023), conditioning on LQ inputs in the latent space. Another set of methods, including DDRM Kawar et al. (2022), CCDF Chung et al. (2022), and DDNM Yang et al. (2021), as well as others Chen et al. (2023a); Csiszár (1975); Wang et al. (2022); Zhang et al. (2023), explore optimization within the latent space by applying controlled degradations to the LQ image and reconstructing HQ outputs. However, these approaches often involve lengthy sampling procedures and are limited by their dependence on fixed degradation models, reducing their flexibility in real-world settings. Despite the progress achieved by these methods, most diffusion-based super-resolution pipelines remain computationally intensive due to their reliance on multi-step sampling procedures, and they often fail to match the level of fine-grained detail produced by comprehensive multi-step diffusion processes.

**One step Real-ISR.** Several methods have attempted to extend multi-step inference-based Real-ISR pipelines to single-step alternatives by incorporating additional refinement strategies. For instance, SinSR Wang et al. (2024b) reduces the four-step ResShift process to a single-step inference by employing a distillation technique that preserves structural information. However, it still falls short in reproducing the fine details typically obtained through multi-step diffusion priors. Similarly, OSEDiff Wu et al. (2024a) leverages variational score distillation as a regularizer to fine-tune SD using LoRA-based adaptation. Another line of work, known as InvSR Yue et al. (2024), focuses on optimizing a set of noise maps that the model can learn to estimate. At inference time, these maps are used to perform the reverse diffusion in a single step. DDIM Song et al. (2020a) was the first method to relax the Markovian assumption inherent in standard diffusion models, enabling the sampling process to incorporate information from both the previous time step and an estimated denoised sample. This reformulation results in a more deterministic and controllable sampling trajectory, thereby significantly improving sampling efficiency and reducing inference time. Building on this, BIRD Chihaoui et al. (2024) further accelerated the DDIM framework by omitting updates to intermediate latent representations during the reverse process, once an initial noise estimate is obtained. Despite these innovations, most of these approaches primarily focus on tweaking internal image representations or conditioning and often fail to deliver perceptually compelling results for high-quality Real-ISR, particularly in scenarios requiring fine structural and texture detail.

## 3 METHOD

### 3.1 ONE-STEP DIFFUSION:

Denoising Diffusion Probabilistic Models (DDPMs) generate high-quality images by modeling a forward diffusion process in which an initial clean image $x_0$ is gradually transformed into pure Gaussian noise $x_T \sim \mathcal{N}(0, \mathrm{I})$ through a sequence of intermediate states $\{x_t\}_{t=1}^T$. This is achieved by

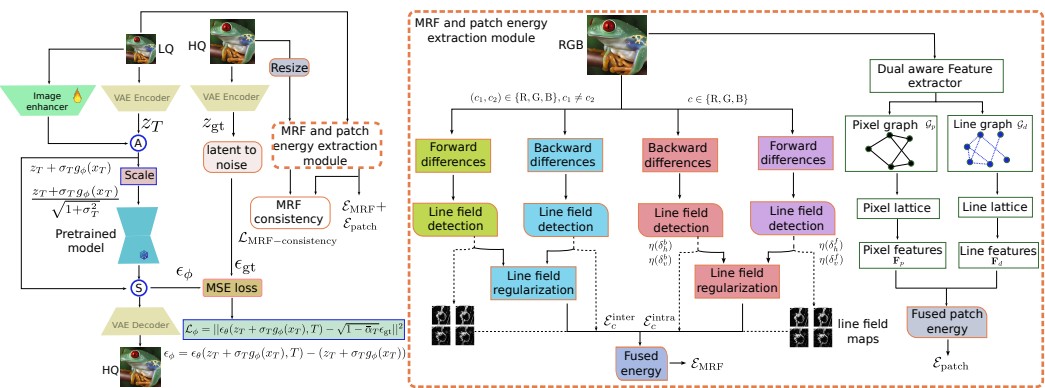

Figure 1: The overall framework of DMAPSR is illustrated as follows: the LQ image is first processed by a VAE encoder and an image enhancer, which together map the image into a latent space tailored for one-step diffusion. The image enhancer is trained to preserve structural discontinuities by minimizing a MAP-optimized MRF energy, complementing the generative capacity of the pretrained SD model. For noise estimation, only the residual noise is learned by the image enhancer, obtained by subtracting the noise predicted by the pretrained SD model. Prior to inputting into SD, both the latent representation from the SD autoencoder and the features from the image enhancer are scaled appropriately to ensure compatibility. Box with the dotted line of the left side is defined elaborately on right side.

incrementally injecting noise at each time step $t \in [1, T]$. To reverse this process and reconstruct the original image, a neural network $\epsilon_\theta$ is trained to predict the noise component at each diffusion step. In our framework, we begin with a pretrained noise prediction model $\epsilon_\theta(x_t, t)$ such as SD, trained via denoising score matching. During training, the model is optimized by minimizing the following objective:

$$\nabla_\theta \|\epsilon - \epsilon_\theta(\sqrt{\bar{\alpha}_t}x_0 + \sqrt{1 - \bar{\alpha}_t}\epsilon, t)\|^2$$

where the noisy sample is defined as: $x_t \simeq \sqrt{\bar{\alpha}_t}x_0 + \sqrt{1 - \bar{\alpha}_t}\epsilon$. From this, we obtain an estimate of the original clean sample $\hat{x}_0$ as, $\hat{x}_0 \simeq \frac{1}{\sqrt{\bar{\alpha}_t}}(x_t - \sqrt{1 - \bar{\alpha}_t}\epsilon)$. Given a pretrained noise predictor $\epsilon_\theta(x_t, t)$, the reverse diffusion step in a typical DDPM is expressed as:

$$x_{t-1} = \frac{1}{\sqrt{\alpha_t}}\left(x_t - \frac{1 - \alpha_t}{\sqrt{1 - \bar{\alpha}_t}}\epsilon_\theta(x_t, t)\right) + \sigma_t z = \mu_\theta(x_t, t) + \sigma_t z \tag{1}$$

Here $z \sim \mathcal{N}(0, I)$ and $\sigma_t^2 = \frac{1 - \bar{\alpha}_{t-1}}{1 - \bar{\alpha}_t}(1 - \alpha_t)$. Here $\mu_\theta(x_t, t)$ represents the predicted mean of the denoised distribution. In contrast, DDIM modify this sampling process to be non-Markovian and potentially deterministic. The DDIM update rule is given by:

$$x_{t-1} = \sqrt{\bar{\alpha}_{t-1}}\left(\frac{x_t - \sqrt{1 - \bar{\alpha}_t}\epsilon_\theta(x_t, t)}{\sqrt{\bar{\alpha}_t}}\right) + \sqrt{1 - \bar{\alpha}_{t-1} - \sigma_t^2}\epsilon_\theta(x_t, t) + \sigma_t z \tag{2}$$

By setting $\sigma_t = 0$, becomes a deterministic process, enabling faster sampling. In this case, the estimate $\hat{x}_0$ and the subsequent update simplifies to:

$$\hat{x}_0 = \frac{x_t - \sqrt{1 - \bar{\alpha}_t}\epsilon_\theta(x_t, t)}{\sqrt{\bar{\alpha}_t}} \quad , \quad x_{t-1} = \sqrt{\bar{\alpha}_{t-1}}\hat{x}_0 + \sqrt{1 - \bar{\alpha}_{t-1}} \cdot \frac{x_t - \sqrt{\bar{\alpha}_t}\hat{x}_0}{\sqrt{1 - \bar{\alpha}_t}} \tag{3}$$

To further accelerate the generation process, we propose a single-step formulation by approximating $x_0$ directly from the final noisy sample $x_T$. Specifically, we define: $x'_0 = x_T + \sigma_T g_\phi(x_T)$ where $g_\phi$ is a learnable image enhancement network that refines the noisy input $x_T$ to approximate the clean sample $x_0$. Rather than predicting the noise $\epsilon$ for multiple diffusion steps, we aim to directly map $x_T$ to a corrected sample $x_0$ such that the pretrained noise predictor $\epsilon_\theta$ accurately estimates the corresponding noise. The final loss used to train $g_\phi$ is defined as:

$$\mathcal{L}_\phi = \|\epsilon - \epsilon_\theta(x_T - \sigma_T g_\phi(x_T), T)\|^2 \tag{4}$$

which ensures that the modified input $x_T + \sigma_T g_\phi(x_T)$ yields a consistent noise estimate under the frozen pretrained model $\epsilon_\theta$. This approach effectively collapses the multi-step DDIM sampling into a single forward pass of $g_\phi$, enabling significantly faster inference while maintaining image fidelity through alignment with the original noise prediction objective.

## 3.2 REAL-ISR AS MAP ESTIMATION:

**Degradation and MAP formulation.** We formulate the real-ISR as Maximum A Posteriori (MAP) estimation, where the goal is to recover the most probable HQ image $\mathbf{X} \in \mathbb{R}^{c \times H_x \times W_x}$ given a LQ observation $\mathbf{Y} \in \mathbb{R}^{c \times H_y \times W_y}$. The MAP estimate seeks to maximize the aposteriori :

$$\hat{\mathbf{X}}_{\text{MAP}} = \arg\max_{\mathbf{X}} \log \mathbb{P}(\mathbf{X}|\mathbf{Y}) = \arg\max_{\mathbf{X}} \log \mathbb{P}(\mathbf{Y}|\mathbf{X}) + \log \mathbb{P}(\mathbf{X}) \tag{5}$$

Here, $H_y = H_x/k, W_y = W_x/k$, where $k \in \{2, 4, 8\}$ and $c \in \{\text{R}, \text{G}, \text{B}\}$. $\mathbb{P}(\mathbf{Y}|\mathbf{X})$ denotes the likelihood and $\mathbb{P}(\mathbf{X})$ denotes the apriori. Real-ISR problem takes the generalized degradation operator Gao & Zhuang (2022) as $\mathbf{Y} = \mathcal{D}_\psi(H(\mathbf{X})) \odot \mathcal{N}$ Where $\mathcal{D}_\psi$ is a parameterized downsampling kernel, $H$ includes aliasing, smoothing, and sparsity priors, and $\mathcal{N} \sim (\mu, \sigma^2)$ denotes additive white Gaussian noise, assumed independent of the underlying Markov structure. The operator $\odot$ denotes element-wise application, which may reduce to addition Gao & Zhuang (2022) in practical scenarios.

**Prior as an MRF: Gibbs Distribution over Pixel-Line Lattice.** We model the image prior $\mathbb{P}(\mathbf{X})$ as a MRF Geman & Geman (1984); Rajagopalan & Chaudhuri (2002) defined over a lattice comprising both image pixel sites and their corresponding dual line sites, with dependencies captured through horizontal and vertical discontinuity fields, referred to as the respective line fields. Let $Z_m$ denote the set of image pixel-sites with each channel of intensity values $\{F_{i,j} = f_{i,j}; (i,j) \in Z_m\}$ denoted as $\{F = f\}$. Here, $\mathcal{F} = \{\mathcal{F}_{i,j}, (i,j) \in Z_m\}$ denote the neighborhood system, where, $\mathcal{F}_{i,j} = \{(k,l); (k,l) \subseteq Z_m\}$ is the neighbour of $(i,j)$. then $\{\mathcal{F}, Z_m\}$ forms an MRF. The full site set is defined as $\mathcal{S} = Z_m \cup D_m$, where $D_m$ represents line-sites capturing spatial transitions. Hence we get an extended neighborhood system $\{\mathcal{G} = \mathcal{G}_s, s \in S\}$, and express the joint prior over both pixel and dual line variables $(f, l)$ as a Gibbs distribution: $\mathbb{P}(F = f, L = l) = \frac{1}{\mathcal{Z}} e^{-\mathcal{E}(f,l)}$, where $\mathcal{E}(f, l) = \sum_{c \in \mathcal{C}} V_c(f, l)$ is the energy function over cliques $\mathcal{C}$ of the graph $\mathcal{G}$, and $V_c$ is the clique potential defined over elements $s \in c$. The partition function $\mathcal{Z}$ ensures proper normalization: $\mathcal{Z} = \sum_{(f,l)} e^{-\mathcal{E}(f,l)}$. Hence the the posterior distribution becomes: $\mathbb{P}(f, l|m) \propto e^{-\mathcal{E}^P(f,l)}$, with $m = \{\mathcal{D}_\psi, H, \mathcal{N}\}$ and corresponding posterior Gibbs distribution over $\{\mathcal{S}, \mathcal{G}^P\}$, with energy,

$$\mathcal{E}^P(f, l) = \mathcal{E}(f, l) + \text{KL}[\mathbb{P}_{\mathbf{Y}|\mathbf{X}}(m|f, l) || \mathbb{P}_{\mathbf{X}}(f, l)] \tag{6}$$

The second term enforces consistency between the prior and the likelihood, analogous to variational inference, by minimizing the KL divergence.

$$\text{KL}[\mathbb{P}_{\mathbf{Y}|\mathbf{X}}(m|f, l) || \mathbb{P}_{\mathbf{X}}(f, l)] = \mathbb{E}_{\mathbf{Y} \sim \mathbb{P}(\mathbf{Y}|\mathbf{X})}[\log \frac{\mathbb{P}_{\mathbf{Y}|\mathbf{X}}(m|f, l)}{\mathbb{P}_{\mathbf{X}}(f, l)}] = \mathbb{E}_{\mathbf{Y}|\mathbf{X}}[\mathcal{E}_{\mathbf{X}}(f, l) - \mathcal{E}_{\mathbf{Y}}(f, l)] + C$$

Where $C = \log \frac{\mathcal{Z}_{\mathbf{X}}}{\mathcal{Z}_{\mathbf{Y}}}$. We interpret this as the MRF consistency loss, which enforces alignment between the prior distribution and the likelihood.

$$\mathcal{L}_{\text{MRF-consistency}} = \mathbb{E}_{\mathbf{Y}|\mathbf{X}}[\mathcal{E}_{\mathbf{X}}(f, l) - \mathcal{E}_{\mathbf{Y}}(f, l)] \tag{7}$$

To regularize the prior, we use a patch energy term that enforces local consistency between pixel sites and the line sites features. Let $\mathbf{F}_p$ and $\mathbf{F}_d$ denote the extracted pixel site and line site features from the pixel lattice and line lattices Geman & Geman (1984); Rajagopalan & Chaudhuri (2002), respectively, and let $\beta(\cdot, \cdot)$ be a concatenation or fusion operator acting on co-located features. The patch energy is defined as,

$$\mathcal{E}_{\text{patch}} = \mathbb{E}_{u \in \Omega_x} \left[ \| \beta(\mathbf{F}_p^{(b)}, \mathbf{F}_d^{(b)})_u \|_2^2 \right] \tag{8}$$

This term promotes spatial coherence by penalizing local feature disparities.

**MRF energy:** To model spatial discontinuities in the image, we define horizontal and vertical difference operators for the $c$-th image channel $\mathbf{X}_c \in \{\mathbf{X}_{\text{R}}, \mathbf{X}_{\text{G}}, \mathbf{X}_{\text{B}}\}$. The horizontal and vertical differences at spatial index $(i, j)$ are given by, $\Delta_h^c(x_{i,j}) = x_{i,j}^c - x_{i,j+1}^c$ and $\Delta_v^c(x_{i,j}) = x_{i,j}^c - x_{i+1,j}^c$. These differences are used to define the line fields—binary discontinuity indicators—for each channel via a hard threshold $\tau$. The horizontal ($v_{i,j}^c$) and vertical ($l_{i,j}^c$) line fields are given as:

$$v_{i,j}^c = \begin{cases} 1, & \text{if } |\Delta_h^c(x_{i,j})| > \tau \\ 0, & \text{otherwise} \end{cases} \quad , \quad l_{i,j}^c = \begin{cases} 1, & \text{if } |\Delta_v^c(x_{i,j})| > \tau \\ 0, & \text{otherwise} \end{cases} \tag{9}$$

The resulting binary fields $\mathbf{L}$ and $\mathbf{V}$ serve as indicators of vertical and horizontal discontinuities, respectively. Gibbs prior distribution over each channel prior combining these line fields as: $\mathbb{P}(\mathbf{X} = x, \mathbf{L} = l, \mathbf{V} = v|c) \propto \exp\{-\mathcal{E}(x^c, l^c, v^c)\}$. The associated energy is given by a first-order weak-membrane energy Rajagopalan & Chaudhuri (2002):

$$\mathcal{E}_{\mathrm{MRF}}(x^c, l^c, v^c) = \sum_{i,j}[(1 - v_{i,j}^c)(\Delta_h^c(x_{i,j}^c))^2 + (1 - v_{i,j-1}^c)(\Delta_h^c(x_{i,j-1}^c))^2$$

$$+(1 - l_{i,j}^c)(\Delta_v^c(x_{i,j}^c))^2 + (1 - l_{i,j-1}^c)(\Delta_v^c(x_{i,j-1}))^2] + \gamma \sum_{i,j}[v_{i,j}^c + v_{i,j-1}^c + l_{i,j}^c + l_{i,j-1}^c] \quad (10)$$

The final term acts as a penalty that discourages the introduction of excessive discontinuities in the recovered image. Setting $\gamma = 0$ eliminates this constraint, leading to a trivial solution in which the MRF tends to introduce discontinuities indiscriminately, including in regions where such transitions are unlikely.

**Discontinuity preservation:** To generalize the MRF energy to RGB images, and enable optimal discontinuity modeling, we replace binary line fields with continuous, differentiable soft line fields: $\eta(\delta) = \sigma(\alpha(|\delta| - \tau))$, where $\sigma(\cdot)$ is the sigmoid function, $\alpha$ controls the sharpness. Let the forward differences across spatial sites be defined as:

$$\delta_h^f(\mathbf{X}_c) = \{\Delta_h^k(x_{i,j})|k \in \{\mathrm{R, G, B}\}, (i,j) \in (0, \cdots H_x) \times (0, \cdots W_x)\}$$
$$\delta_v^f(\mathbf{X}_c) = \{\Delta_v^k(x_{i,j})|k \in \{\mathrm{R, G, B}\}, (i,j) \in (0, \cdots H_x) \times (0, \cdots W_x)\}$$

with backward differences defined as, $\delta_h^b = -\delta_h^f$ and $\delta_v^b = -\delta_v^f$. The intra-channel compound MRF energy for channel $c$ is given by:

$$\mathcal{E}_c^{\mathrm{intra}} = \mathbb{E}_{\mathrm{spatial}}[(1 - \eta(\delta_h^f)) \cdot (\delta_h^f)^2 + (1 - \eta(\delta_v^f)) \cdot (\delta_v^f)^2] + \mathbb{E}_{\mathrm{spatial}}[(1 - \eta(\delta_h^b)) \cdot (\delta_h^b)^2$$
$$+(1 - \eta(\delta_v^b)) \cdot (\delta_v^b)^2] + \gamma\bar{\eta} \quad (11)$$

where $\bar{\eta}$ is the mean soft line field value. Inter-channel energy terms $\mathcal{E}_{c_1,c_2}^{\mathrm{inter}}$ are similarly computed using differences between different color channels. The overall MRF energy across all channels is then:

$$\mathcal{E}_{\mathrm{MRF}}(\mathbf{X}) = \frac{1}{3}\sum_{c \in \{\mathrm{R,G,B}\}} \mathcal{E}_c^{\mathrm{intra}} + \frac{1}{3}\sum_{\substack{(c_1,c_2) \in \{\mathrm{R,G,B}\} \\ c_1 \neq c_2}} \mathcal{E}_{c_1,c2}^{\mathrm{inter}} \quad (12)$$

Combining 4,6, 7 and 8, the final objective function becomes:

$$\mathcal{L} = \mathcal{L}_\phi + \mathcal{E}_{\mathrm{MRF}} + \lambda_p\mathcal{E}_{\mathrm{patch}} + \lambda_r\mathcal{L}_{\mathrm{MRF-consistency}} \quad (13)$$

Overall architecture is shown in Fig. 1.

# 4 EXPERIMENTS

EXPERIMENTAL DETAILS:

**Training and testing dataset:** Previous works Wang et al. (2024a); Lin et al. (2024); Wu et al. (2024b); Yue et al. (2023) have employed a variety of datasets for the $\times 4$ Real-ISR task. In line with Wu et al. (2024b;a); Yue et al. (2024), we use the LSDIR Li et al. (2023) dataset and the first 10,000 face images from FFHQ Karras et al. (2019) for training our model. LQ images are generated using the degradation pipeline proposed in Real-ESRGAN Wang et al. (2021). The model is trained using the Adam optimizer with a batch size of 64 for 90,000 iterations and a fixed learning rate of 5e-5. We evaluate DMAPSR on both real-world datasets, including RealSR Cai et al. (2019) and DRealSR Wei et al. (2020), as well as on the synthetic DIV2K validation set Agustsson & Timofte (2017). The hyperparameters $\lambda_p$, $\lambda_r$, and $\gamma$ are set to 1, 1, and 0.1, respectively.

**Compared Methods.** We compare our proposed method against a range of state-of-the-art approaches, including GAN-based BSRGAN Zhang et al. (2021), as well as diffusion-based methods such as StableSR Wang et al. (2024a), DiffBIR Lin et al. (2024), SeeSR Wu et al. (2024b), ResShift Yue et al. (2023), SinSR Wang et al. (2024b), OSEDiff Wu et al. (2024a), InvSR Yue et al.

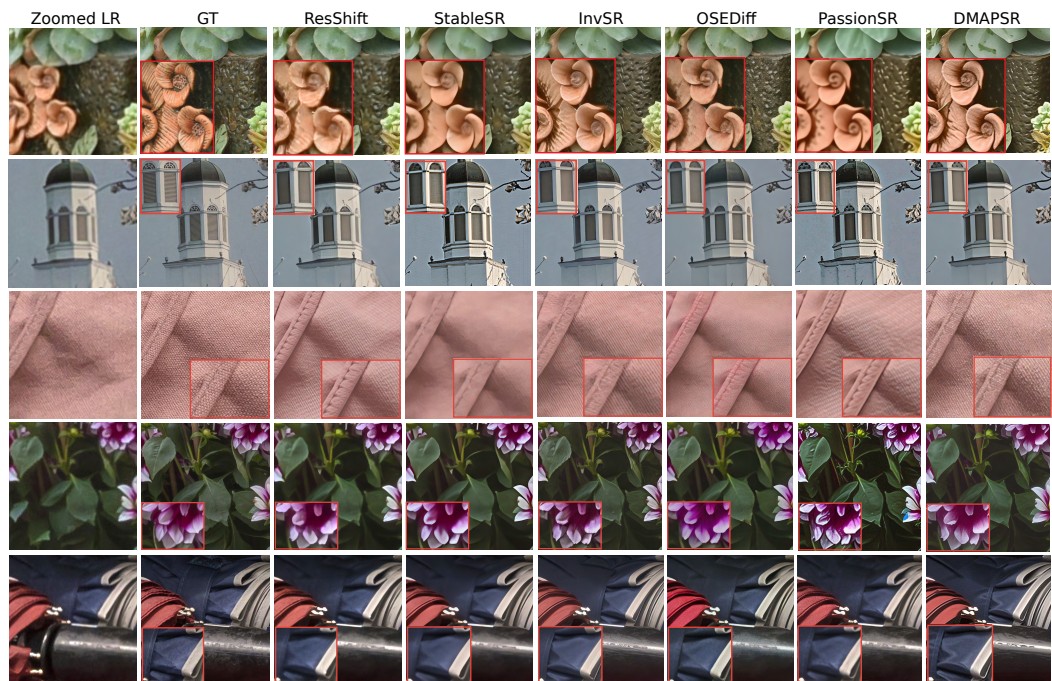

Figure 2: Qualitative visual comparisons of Real-ISR methods. Please zoom in for a better view.

| Datasets | Methods | Evaluation Metrics | | | | | | | | |
|---|---|---|---|---|---|---|---|---|---|---|
| | | PSNR ↑ | SSIM ↑ | LPIPS ↓ | MUSIQ ↑ | CLIPIQA ↑ | DISTS ↓ | FID ↓ | NIQE ↓ | MANIQA ↑ |
| DIV2K-Val | BSRGAN | 21.87 | 0.5539 | 0.4136 | 59.11 | 0.5183 | 0.2737 | 64.28 | 4.7615 | 0.4834 |
| | DiffBIR-50 | 23.64 | 0.5647 | 0.3524 | 65.81 | 0.6704 | 0.2128 | 30.72 | 4.7042 | 0.6210 |
| | StableSR-50 | 23.26 | 0.5726 | 0.3113 | 65.92 | 0.6771 | 0.2048 | **24.44** | 4.7581 | 0.6192 |
| | SeeSR-50 | 23.68 | 0.6043 | 0.3194 | **68.67** | **0.6936** | 0.1968 | 25.90 | 4.8102 | 0.6240 |
| | ResShift-4 | **24.65** | **0.6181** | 0.3349 | 61.09 | 0.6071 | 0.2213 | 36.11 | 6.8212 | 0.5454 |
| | SinSR-1 | 24.41 | 0.6018 | 0.3240 | 62.82 | 0.6471 | 0.2066 | 35.57 | 6.0159 | 0.5386 |
| | OSEDiff-1 | 23.72 | 0.6108 | 0.2941 | 67.97 | 0.6683 | 0.1976 | 26.32 | 4.7097 | 0.6148 |
| | PassionSR-1 | 24.34 | 0.7097 | 0.3440 | 51.19 | 0.4802 | 0.2075 | 28.45 | 7.039 | 0.2267 |
| | DMAPSR | 24.58 | 0.6112 | **0.2938** | 68.52 | 0.6842 | **0.1962** | 27.19 | **4.6721** | **0.6416** |
| DRealSR | BSRGAN | 28.75 | 0.8031 | **0.2883** | 57.14 | 0.4915 | 0.2142 | 155.63 | 6.5192 | 0.4878 |
| | DiffBIR-50 | 26.71 | 0.6571 | 0.4557 | 61.07 | 0.6395 | 0.2748 | 166.79 | **6.3124** | 0.5930 |
| | StableSR-50 | 28.03 | 0.7536 | 0.3284 | 58.51 | 0.6356 | 0.2269 | 148.98 | 6.5239 | 0.5601 |
| | SeeSR-50 | 28.17 | 0.7691 | 0.3189 | 64.93 | 0.6804 | 0.2315 | 147.39 | 6.3967 | 0.6042 |
| | ResShift-4 | **28.46** | 0.7673 | 0.4006 | 50.60 | 0.5342 | 0.2656 | 172.26 | 8.1249 | 0.4586 |
| | SinSR-1 | 28.36 | 0.7515 | 0.3665 | 55.33 | 0.6383 | 0.2485 | 170.57 | 6.9907 | 0.4884 |
| | OSEDiff-1 | 27.92 | 0.7835 | 0.2968 | 64.65 | 0.6963 | 0.2165 | **135.30** | 6.4902 | 0.5899 |
| | DMAPSR-1 | 28.32 | **0.7842** | 0.2957 | **64.97** | **0.6975** | **0.2096** | 139.75 | 6.3245 | **0.6172** |
| RealSR | BSRGAN | 26.39 | 0.7654 | 0.2670 | 63.21 | 0.5001 | 0.2121 | 141.28 | 5.6567 | 0.5399 |
| | DiffBIR-50 | 24.75 | 0.6567 | 0.3636 | 64.98 | 0.6463 | 0.2312 | 128.99 | 5.5346 | 0.6246 |
| | StableSR-50 | 24.70 | 0.7085 | 0.3018 | 65.78 | 0.6178 | 0.2288 | 128.51 | 5.9122 | 0.6221 |
| | SeeSR-50 | 25.18 | 0.7216 | 0.3009 | 69.77 | 0.6612 | 0.2223 | 125.55 | **5.4081** | 0.6442 |
| | ResShift-4 | **26.31** | 0.7421 | 0.3460 | 58.43 | 0.5444 | 0.2498 | 141.71 | 7.2635 | 0.5285 |
| | SinSR-1 | 26.28 | 0.7347 | 0.3188 | 60.80 | 0.6122 | 0.2353 | 135.93 | 6.2872 | 0.5385 |
| | OSEDiff-1 | 25.15 | 0.7341 | 0.2921 | 69.09 | **0.6693** | **0.2128** | **123.49** | 5.6476 | 0.6326 |
| | PassionSR-1 | 22.52 | 0.6255 | 0.4913 | 43.21 | 0.3089 | 0.3185 | 129.54 | 5.706 | 0.2396 |
| | DMAPSR-1 | 26.29 | **0.7426** | **0.2918** | 69.81 | 0.6651 | 0.2178 | 127.92 | 5.4104 | **0.6492** |

Table 1: We conduct a quantitative comparison of DMAPSR with state-of-the-art Real-ISR models based on GAN and diffusion frameworks across various datasets with the reverse timestep after hyphen. The best-performing method is highlighted in bold, while the second-best result is indicated with an underline.

(2024), and PassionSR Zhu et al. (2024). For a fair comparison, we follow the official configurations of each method. StableSR, DiffBIR, and SeeSR are evaluated using 50 sampling steps, as originally proposed. ResShift is evaluated using 4 sampling steps, while SinSR, OSEDiff, PassionSR, and InvSR are all evaluated using a single sampling step, in accordance with their respective official implementations.

| Metrics | StableSR | DiffBIR | SeeSR | ResShift | SinSR | OSEDiff | InvSR | DMAPSR |
|---|---|---|---|---|---|---|---|---|
| Inference step | 50 | 50 | 50 | 4 | 1 | 1 | 1 | 1 |
| Inference time (s) | 11.50 | 2.72 | 4.29 | 0.71 | 0.13 | 0.15 | 0.12 | 0.11 |
| #Total Params(M) | 1410 | 1717 | 2524 | 119 | 119 | 1775 | 1145 | 949 |
| #Trainable Params(M) | 150 | 380 | 750 | 119 | 119 | 8.5 | 33.84 | 33.51 |

Table 2: Comparison of inference time and parameter count across different methods. All evaluations are conducted on a single NVIDIA A-100 GPU with a maximum memory capacity of 80GB, for the $\times 4(128 \rightarrow 512)$ SR task.

COMPARISON TO THE STATE OF THE ART:

**Quantitative Comparisons.** The quantitative comparison across three benchmarks is presented in Table 1. In full-reference image quality assessment metrics, DMAPSR demonstrates superior performance over existing methods, achieving the best or second-best scores in SSIM and the perceptual quality metric LPIPS on both the RealSR and DRealSR benchmarks. Additionally, in the structural similarity metric DISTS, DMAPSR consistently performs well across all benchmarks. In terms of semantic and content-aware evaluation, CLIPIQA scores indicate that DMAPSR outperforms all competing methods on all three datasets. For no-reference IQA metrics, while SeeSR and OSEDiff exhibit strong performance, DMAPSR achieves comparable or better results in perceptual quality metrics such as MUSIQ and MANIQA. In the FID score, SeeSR performs better due to the advantage of multi-step generation in capturing global content alignment. ResShift yields strong results in the pixel-wise PSNR metric, benefiting from end-to-end training from scratch, which facilitates better alignment with pixel-level fidelity. However, it underperforms in perceptual and content-based metrics. Overall, DMAPSR achieves leading performance among methods based on pretrained SD priors, especially in the single-step inference setting, demonstrating a favorable trade-off between efficiency and perceptual quality.

**Qualitative Comparisons.** Fig. 2 presents qualitative comparisons of our method against several existing approaches. In the first example, ResShift, which is trained from scratch without leveraging SD priors, produces a slightly blurred facial region with reduced detail. Similarly, the prior-based StableSR exhibits some blur, indicating limitations in capturing fine textures. PassionSR consistently produces over-sharpened and over-brightened outputs across examples, likely due to post-training quantization effects. While InvSR and OSEDiff benefit from SD priors and generally perform well, they tend to introduce unnatural details, particularly noticeable in the third example, thereby deviating from the ground truth.

In contrast, DMAPSR, despite being a single-step diffusion method, produces visually faithful reconstructions that are both texture-rich and aligned with the natural properties of the original image across all examples. Notably, in the third and fifth examples, OSEDiff fails to reconstruct fine textures, underscoring the limitations of approaches that depend exclusively on prompt-based supervision. Furthermore, while OSEDiff requires text prompts during training, it struggles to maintain reconstruction quality during inference in the absence of such external guidance. Overall, DMAPSR demonstrates the ability to generate

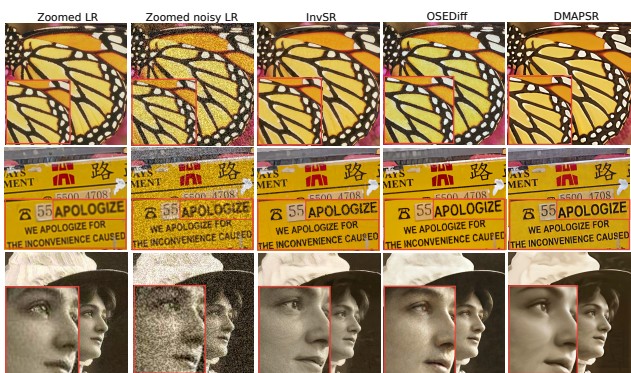

Figure 3: Qualitative comparison of one-step diffusion methods under the setting where the LQ input is further degraded with additive noise. Please zoom in for a better view.

natural, high-fidelity results without relying on prompt-based training, offering a significant advantage in generating realistic textures in a single-step inference setting. More qualitative results are provided in the supplementary.

**Runtime and computational overhead.** Table 2 presents the runtime performance and computational overhead of our method compared to existing approaches, evaluated on a single NVIDIA A-100 GPU using $512 \times 512$ images from the DRealSR benchmark. Among all one-step diffusion models, DMAPSR achieves the second lowest trainable parameter count and the fastest inference speed, while

| Methods | PSNR ↑ | LPIPS ↓ | MUSIQ ↑ | CLIPIQA ↑ |
|---|---|---|---|---|
| i.w/o consist. | 26.07 | 0.2959 | 69.21 | 0.6581 |
| ii.w/o patch | 26.04 | 0.2947 | 69.15 | 0.6581 |
| iii.MRF-1C | 25.96 | 0.3046 | 68.92 | 0.6542 |
| iv.w/o inter | 26.14 | 0.2963 | 69.62 | 0.6627 |
| v.DMAPSR | 26.29 | 0.2918 | 69.81 | 0.6651 |

| Methods | PSNR ↑ | LPIPS ↓ | MUSIQ ↑ | CLIPIQA ↑ |
|---|---|---|---|---|
| ConvNeXtLiu et al. (2022) | 25.97 | 0.3189 | 68.96 | 0.6541 |
| RestormerZamir et al. (2021) | 25.44 | 0.3147 | 68.52 | 0.6581 |
| ResUNetDiakogiannis et al. (2020) | 25.26 | 0.3226 | 68.33 | 0.6522 |
| NAFNetChen et al. (2022b) | 25.68 | 0.3043 | 68.31 | 0.6467 |
| VQGAN-EEsser et al. (2020) | 26.29 | 0.2918 | 69.81 | 0.6651 |

Table 3: Comparison of different losses on RealSR dataset.

Table 4: Ablation on the image enhancer network on the RealSR dataset.

also outperforming multi-step methods. Specifically, DMAPSR provides nearly $100\times$ faster inference than the multi-step StableSR, while requiring only one-fifth the number of trainable parameters. It is also approximately $6\times$ faster than ResShift and $1.3\times$ faster than the single-step OSEDiff, all while maintaining superior output quality. Although OSEDiff has the smallest number of trainable parameters, DMAPSR significantly reduces the overall parameter count—almost by half—making it more suitable for deployment scenarios due to its compactness and higher efficiency.

**Noise removal.** In the LQ image, we introduce additive noise to further challenge the reconstruction process. We evaluate the noise removal capability of our method alongside other approaches, as illustrated in Fig. 3. This evaluation highlights the effectiveness of our model in denoising. The results demonstrate that our method outperforms other one-step diffusion frameworks, such as OSEDiff and InvSR, both of which exhibit notable degradation in reconstruction quality under noisy conditions.

ABLATION EXPERIMENTS:

**Loss function components.** To assess the contribution of various loss terms in our framework, we conduct an ablation study on the RealSR benchmark, with results presented in Table 3. Specifically, we evaluate the performance under the following settings: (i) removing the MRF-consistency term, (ii) excluding the patch-based energy term, (iii) computing MRF energy on a single grayscale channel instead of full RGB, and (iv) omitting the inter-channel MRF energy term. These are compared against the full model that incorporates all components of Equation 13. The absence of either the patch-based energy term or the MRF-consistency term leads to a notable decline in reference-based PSNR. This indicates that both terms are essential for preserving fine-grained details in the LQ image and maintaining accurate correspondence with the ground truth. When MRF energy is computed solely on a grayscale channel, the model fails to capture the diverse local interactions present across the RGB channels, resulting in degraded performance. Furthermore, excluding the inter-channel component of the MRF energy significantly impairs the model's ability to reconstruct rich textures and color details, demonstrating its importance in modeling cross-channel dependencies. In addition to that we also show some qualitative results for this in Fig. 4.

In addition to the MRF loss, we incorporate the MRF-consistency loss and the patch-energy loss to train our image enhancer, as defined in Eq. (13). We also include a penalty term, $\gamma$, which regulates the number of discontinuities, preventing them from becoming excessively sparse or overly dense. The hyperparameters $\lambda_p$ and $\lambda_r$ control the relative contributions of the patch-energy and MRF-consistency losses, respectively. Table 5 presents a quantitative comparison across different loss configurations, and Fig.5 illustrates a representative visual result. As observed, a large value of $\gamma$ leads to oversmoothing, whereas a small value encourages the insertion of excessive structural lines. Furthermore, an appropriate balance between the patch-energy and MRF-consistency terms enhances local contrast, improves overall image quality, and suppresses artifacts. These components are therefore essential for achieving optimal performance.

**Image enhancer architecture.** We conduct an ablation study to investigate the effect of different backbone architectures for the image enhancer module in our DMAPSR framework. The results, reported in Table 4, are evaluated on the RealSR benchmark using both perceptual and fidelity-based metrics with five backbone variants. Among these, the VQGAN Esser et al. (2020) encoder-based design achieves the best trade-off, yielding the highest PSNR. This superior performance highlights the importance of preserving both structural and semantic features during latent-space transformation. The VQGAN-style downsampling blocks are particularly effective in capturing localized texture and long-range dependencies while compressing the image representation, making them well-suited for our one-step diffusion framework. Therefore, we adopt the VQGAN-based architecture as the default image enhancer in our pipeline.

Table 5: Quantitative ablation studies on the loss function in Eq. (11), wherein the hyper-parameters $\lambda_p$, $\lambda_r$ control the weight importance of the patch energy and the MRF-consistency, respectively and the $\gamma$ controls the MRF-penalty.

| Hyper-parameters | | | Metrics | | | | | | | |
|---|---|---|---|---|---|---|---|---|---|---|
| $\lambda_p$ | $\lambda_r$ | $\gamma$ | PSNR ↑ | LPIPS ↓ | MUSIQ ↑ | CLIPIQA ↑ | SSIM ↑ | DISTS ↓ | NIQE ↓ | NIQE ↑ |
| 0.2 | 0.2 | 0.3 | 27.92 | 0.3317 | 61.67 | 0.6521 | 0.7718 | 0.2502 | 7.52 | 0.5715 |
| 2.0 | 0.5 | 0.05 | 27.15 | 0.3254 | 62.82 | 0.6854 | 0.7671 | 0.2305 | 7.14 | 0.5891 |
| 0.5 | 2.0 | 0.2 | 28.82 | 0.3154 | 61.87 | 0.6621 | 0.7861 | 0.2215 | 7.82 | 0.5821 |
| 0.5 | 0.5 | 0.1 | 28.15 | 0.3125 | 62.18 | 0.6772 | 0.7642 | 0.2271 | 6.85 | 0.6072 |
| 1.0 | 1.0 | 0.1 | 28.32 | 0.2957 | 64.97 | 0.6975 | 0.7842 | 0.2096 | 6.3245 | 0.6172 |

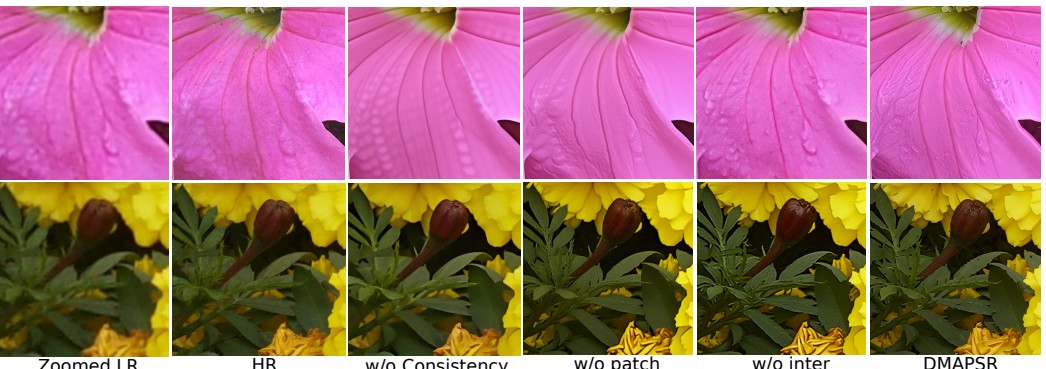

|        |       |                |           |           |        |
|--------|-------|----------------|-----------|-----------|--------|
| Zoomed LR | HR | w/o Consistency | w/o patch | w/o inter | DMAPSR |

Figure 4: Qualitative visual comparisons of Real-ISR methods with different confifurations of MRF settings and the patch energy. Please zoom in for a better view.

**Extension to Multi-Step Formulation.** In addition to the single-step variant, we also provide a multi-step formulation of DMAPSR. The details of this extended version are included in the Supplementary Material.

## 5 CONCLUSION

We propose DMAPSR, a single-step diffusion-based super-resolution method that explicitly preserves image discontinuities while enabling efficient reconstruction. Our approach combines a lightweight image enhancer with a pretrained diffusion backbone for structure-aware detail synthesis and robust noise estimation. DMAPSR introduces a discontinuity-preserving line field energy, optimized via a MRF formulation, which ensures the reconstruction of sharp structural edges. The model further captures fine-grained information within and across RGB channels, enabling enhanced texture fidelity alongside rapid sampling. Experimental results demonstrate that DMAPSR achieves comparable or superior performance to both single-step and multi-step real-image super-resolution baselines in terms of objective quality metrics and visual fidelity.

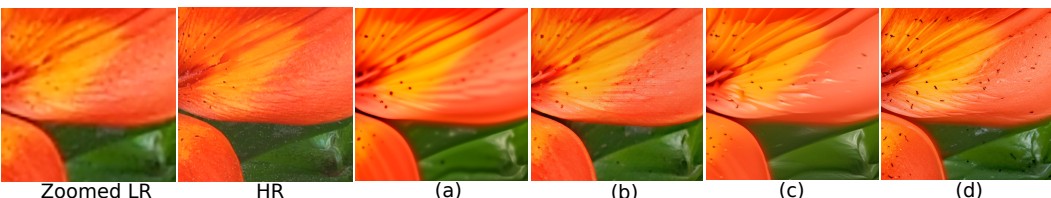

|        |       |     |     |     |     |
|--------|-------|-----|-----|-----|-----|
| Zoomed LR | HR | (a) | (b) | (c) | (d) |

Figure 5: Qualitative visual comparisons of Real-ISR methods with different hypeparameter values. (a)$\lambda_r = 0.2, \lambda_p = 0.2, \gamma = 0.05$, (b)$\lambda_r = 2.0, \lambda_p = 0.5, \gamma = 0.3$, (c)$\lambda_r = 0.5, \lambda_p = 2.0, \gamma = 0.2$, (d)$\lambda_r = 1.0 \lambda_p = 1.0, \gamma = 0.1$ Please zoom in for a better view.

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

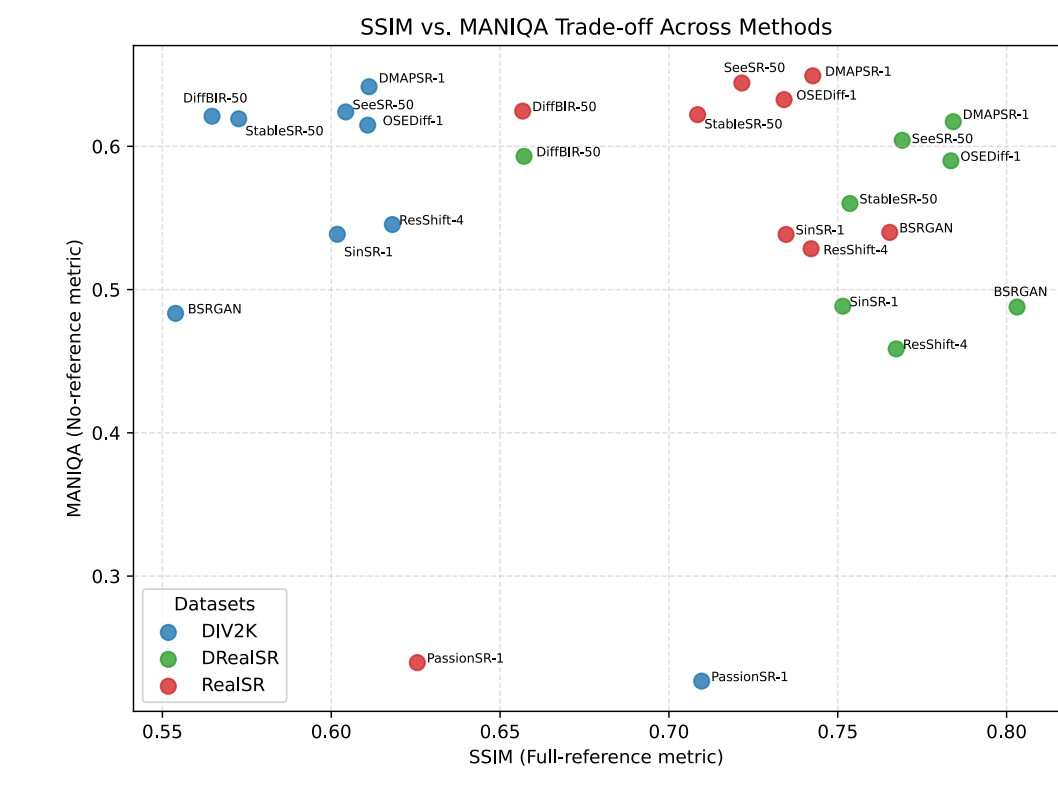

Figure 6: Scatter plot illustrating model performance, where SSIM (full-reference metric) is shown on the x-axis and MANIQA (no-reference metric) on the y-axis. This visualization enables a fair comparison across methods and clearly highlights the trade-off characteristics between reference-based fidelity and perceptual quality.

## A SUPPLEMENTARY MATERIAL

In the supplementary material, we provide the following additional details:

- Scatter plot illustrating model performance in Fig.6 and Fig.7.
- A complete proof of the posterior energy formulation presented in the main paper.
- Visualizations of the line fields along with the corresponding images.
- Additional qualitative and quantitative results to further support our findings.
- Usage of LLM

### PROOF OF THE MRF ENERGY OF THE POSTERIOR:

**Theorem:** Let the prior distribution $\mathbb{P}(X = \omega)$ be a Gibbs distribution defined over a neighborhood $\{S, \mathcal{G}\}$ with corresponding energy $\mathcal{E}$ and potential$\{V_C\}$: $\mathbb{P}(X = \omega) = e^{-\mathcal{E}(\omega)}/Z$, $\mathcal{E}(\omega) = \sum_C V_C(\omega)$, where $\omega = (f, l)$. Then, for any fixed observation $g$, the posterior distribution $\mathbb{P}(X = \omega | G = g)$ is also a Gibbs distribution, defined over the neighborhood system $\{S, \mathcal{G}^P\}$, with the posterior energy function given by:

$$\mathcal{E}^P(f, l) = \mathcal{E}(f, l) + \frac{1}{2\sigma^2}||\mu - \Phi(g, \mathcal{D}_\psi(H(F)))||^2 \tag{14}$$

where $\mathcal{G}^P$ denotes the extended neighborhood system defined as:

$$\mathcal{G}_s^P = \begin{cases} \mathcal{G}_s, & \text{if } s \in D_m \\ \mathcal{G}_s \cup \mathcal{H}_s^2 \setminus \{s\}, & \text{if } s \in Z_m \end{cases} \tag{15}$$

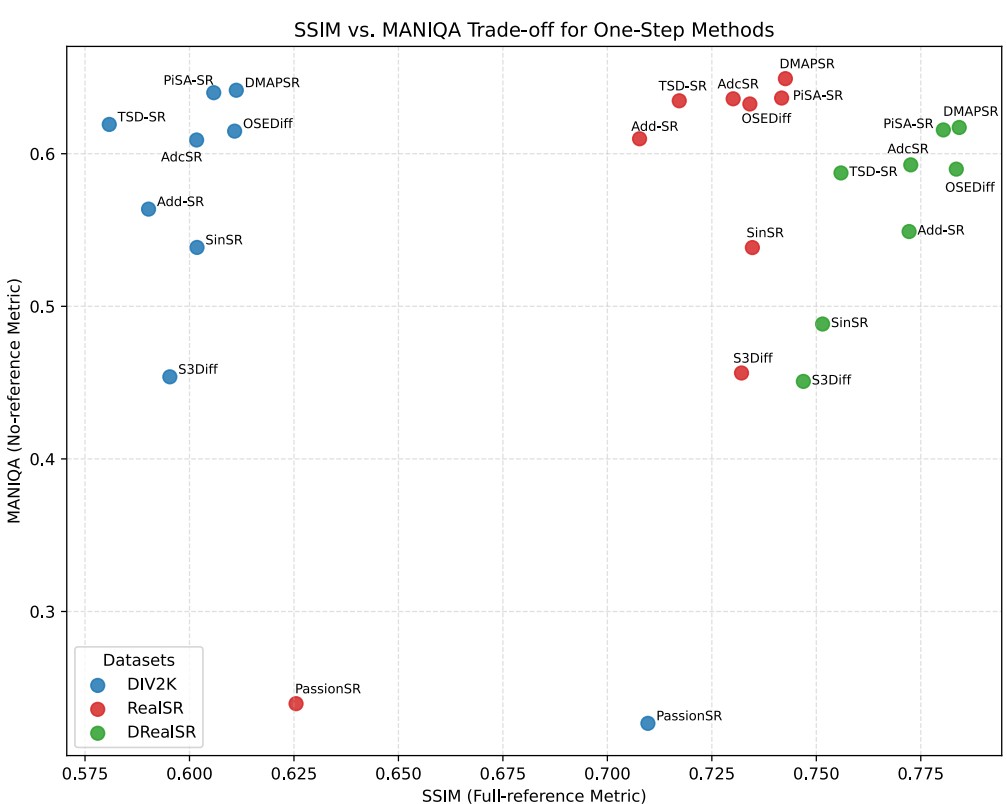

Figure 7: Scatter plot illustrating model performance of one step methods, where SSIM (full-reference metric) is shown on the x-axis and MANIQA (no-reference metric) on the y-axis. This visualization enables a fair comparison across methods and clearly highlights the trade-off characteristics between reference-based fidelity and perceptual quality.

*proof:* We start with the definition of the degradation operator, $G = \phi(H(F)) \odot \mathcal{N}$, where $\mathcal{N} \sim \mathcal{N}(\mu, \sigma^2)$ is additive white Gaussian noise, assumed independent of the MRF $\{S, \mathcal{G}\}$. The operation $\odot$ is assumed to be invertible, such that we can write $\mathcal{N} = \Phi(G, \mathcal{D}_\psi(H(F))) = \{\Phi_s, s \in Z_m\}$. Let $\mathcal{H}_s, s \in Z_m$ denote the set of pixels that affect the blurred image $H(F)$ at $s$. For instance $\mathcal{H}_s$ can be a $3 \times 3$ window centered at $s$. $\{\Phi_s, s \in Z_m\}$ depends only on $g_s$ and $\{f_t : t \in \mathcal{H}_s\}$. Because of the shift invariance of $H$, these neighborhoods satisfy $\mathcal{H}_{r+s} = s + \mathcal{H}_r$ where $\mathcal{H}_r \subseteq Z_m, s + r \in Z_m$ and $s + \mathcal{H}_r$ intersects $Z_m$. If $\{\mathcal{H}_s\}$ is symmetric such that $r \in \mathcal{H}_0 \implies -r \in \mathcal{H}_0$, then the collection $\{\mathcal{H}_s \backslash \{s\}, s \in Z_m\}$ forms a valid neighborhood system over $Z_m$. Let $\mathcal{H}^2$ define the second-order neighborhood system as:

$$\mathcal{H}_s^2 = \cup_{r \in \mathcal{H}_s} \mathcal{H}_r, s \in Z_m \tag{16}$$

Then $\{\mathcal{H}_s^2 \backslash \{s\}, s \in Z_m\}$ also defines a neighborhood system. We define the posterior neighborhood system $\{\mathcal{G}^P = \mathcal{G}_s^P, s \in S\}$ as,

$$\mathcal{G}_s^P = \begin{cases} \mathcal{G}_s, & \text{if } s \in D_m \\ \mathcal{G}_s \cup \mathcal{H}_s^2 \backslash \{s\}, & \text{if } s \in Z_m \end{cases} \tag{17}$$

Applying Bayes' rule, we express the posterior as:

$$\mathbb{P}(X = \omega | G = g) = \frac{\mathbb{P}(G = g | X = \omega) \cdot \mathbb{P}(X = \omega)}{\mathbb{P}(G = g)} \tag{18}$$

$\forall \omega = (f, l)$ and each $g$. Assuming $\mathbb{P}(X = \omega) = e^{-\mathcal{E}(\omega)}/Z$, the likelihood term becomes:

$$\begin{aligned} \mathbb{P}(G = g | X = \omega) &= \mathbb{P}(\mathcal{D}_\psi(H(X)) \odot \mathcal{N} = g | X = \omega) \\ &= \mathbb{P}(\mathcal{N} = \Phi(g, \mathcal{D}_\psi(H(X)))) \\ &= (2\pi\sigma^2)^{-M/2} \exp\{-\frac{1}{2\sigma^2}||\mu - \Phi||^2\} \end{aligned} \tag{19}$$

Again, $\mathbb{P}(X = \omega | G = g) = e^{-\mathcal{E}^P(\omega)}/Z^P$.

**Case** $s \in Z_m$: The term $\Phi$ does not cancel out. $\Phi(g, \mathcal{D}_\psi(H(F))) = \{\Phi_s, s \in Z_m\}$.

Taking Eq 18 and Eq. 19 we can write,

$$\mathbb{P}(X = \omega \mid \mathcal{N} = \Phi) \propto \mathbb{P}(\mathcal{N} = \Phi \mid X = \omega) \cdot \mathbb{P}(X = \omega). \tag{20}$$

Taking the negative logarithm, the posterior energy becomes:

$$\mathcal{E}^P(f, l) = -\log \mathbb{P}(\mathcal{N} = \Phi \mid X = \omega) - \log \mathbb{P}(X = \omega). \tag{21}$$

From Eq. 19 we get,

$$-\log \mathbb{P}(\mathcal{N} = \Phi \mid X = \omega) = \frac{1}{2\sigma^2} \sum_{r \in Z_m} (\Phi_r(g_r; f_t, t \in \mathcal{H}_r) - \mu)^2 + \text{const.} \tag{22}$$

Combining both terms, the full posterior energy becomes:

$$\mathcal{E}^P(f, l) = \sum_C V_C(f, l) + \frac{1}{2\sigma^2} \sum_{r \in Z_m} (\Phi_r(g_r; f_t, t \in \mathcal{H}_r) - \mu)^2. \tag{23}$$

$$\mathbb{P}(F_s = f_s \mid F_r = f_r, r \neq s, r \in Z_m, L = l, G = g) = \frac{e^{-\mathcal{E}^P(\omega)}/Z^P}{\sum_{f_s} e^{-\mathcal{E}^P(\omega)}/Z^P} = \frac{e^{-\mathcal{E}(\omega)}/Z}{\sum_{f_s} e^{-\mathcal{E}(\omega)}/Z}$$

$$= \frac{e^{-\mathcal{E}(f,l) - \frac{1}{2\sigma^2} \sum_{r \in Z_m} (\Phi_r - \mu)^2}}{\sum_{f_s} e^{-\mathcal{E}(f,l) - \frac{1}{2\sigma^2} \sum_{r \in Z_m} (\Phi_r - \mu)^2}} \tag{24}$$

$$\begin{aligned} \Rightarrow \quad \mathcal{E}^P(f, l) = &\sum_{C:s \in C} V_C(f, l) + \frac{1}{2\sigma^2} \sum_{r:s \in \mathcal{H}_r} (\Phi_r(g_r; f_t, t \in \mathcal{H}_r) - \mu)^2 \\ &+ \sum_{C:s \notin C} V_C(f, l) + \frac{1}{2\sigma^2} \sum_{r:s \notin \mathcal{H}_r} (\Phi_r(g_r; f_t, t \in \mathcal{H}_r) - \mu)^2 \end{aligned} \tag{25}$$

It can be seen that the last two terms in 25 does not involve $f_s$ and the ratio in 24 depends only on the first two terms of 25. The first two terms depends only on the coordinate $(f, l)$ for the sites in $\mathcal{G}_s\{s \in C \implies C \subseteq \mathcal{G}_s\}$ and the second term only on the sites in $= \cup_{r:s\in\mathcal{H}_r}\mathcal{H}_s = \cup_{r\in\mathcal{H}_s}\mathcal{H}_r = \mathcal{H}_s^2$. Hence we can say, $\mathcal{G}_s^P = \mathcal{G}_s \cup \mathcal{H}_s^2 \backslash \{s\}$.

**Case $s \in D_m$:**

$$\mathbb{P}(L_s = l_s | L_r = l_r, r \neq s, r \in D_m, F = f, G = g)$$

$$= \frac{e^{-\mathcal{E}^P(\omega)}/Z^P}{\sum_{l_s} e^{-\mathcal{E}^P(\omega)}/Z^P} = \frac{e^{-\mathcal{E}(\omega)}/Z}{\sum_{l_s} e^{-\mathcal{E}(\omega)}/Z}$$

The sum extends over all possible values of $L_s$ Hence we can say, $\mathcal{G}_S^P = \mathcal{G}_S$.

Thus, the posterior energy becomes,

$$\mathcal{E}^P(f, l) = \mathcal{E}(f, l) + \frac{1}{2\sigma^2}||\mu - \Phi(g, \mathcal{D}_\psi(H(F)))||^2 \tag{26}$$

**Corollary:** It can be observed that the second term is strictly positive. To generalize this further, we note that this term can be interpreted as a discrepancy measure between the likelihood and the prior. While the KL divergence is a common choice—being strictly positive—other discrepancy measures may also be employed. To demonstrate the similarity between the second term and the KL divergence, we proceed as follows:

$$\mathbb{P}(G = g | X = \omega) = \frac{1}{(2\pi\sigma^2)^{M/2}} \exp\{-\frac{1}{2\sigma^2}||\mu - \Phi(g, \mathcal{D}_\psi(H(F)))||^2\}$$

$$\implies \log \mathbb{P}(G = g | X = \omega) = -\frac{1}{2\sigma^2}||\mu - \Phi(g, \mathcal{D}_\psi(H(F)))||^2 + C$$

Where $C = -\frac{M}{2}\log(2\pi\sigma^2)$ as $X$ is independent of $\mathcal{N}$. Now, taking $\langle h(X) \rangle = E_X[h(X)]$ we can write the above as,

$$\text{KL}[\mathbb{P}(X = \omega)||\mathbb{P}(G = g | X = \omega)] = \langle \log \mathbb{P}(X = \omega) \rangle - \langle \log \mathbb{P}(G = g | X = \omega) \rangle$$

$$= \frac{1}{2\sigma^2}\langle ||\mu - \Phi(g, \mathcal{D}_\psi(H(F)))||^2 \rangle + C - \langle -\frac{U(\omega)}{Z} \rangle$$

$$= \frac{1}{2\sigma^2}\langle ||\mu - \Phi(g, \mathcal{D}_\psi(H(F)))||^2 \rangle + C$$

Hence, effectively we can write the posterior energy as,

$$\mathcal{E}^P(f, l) = \mathcal{E}(f, l) + \text{KL}[\mathbb{P}_{\mathbf{Y}|\mathbf{X}}(g|f,l)||\mathbb{P}_{\mathbf{X}}(f,l)] \tag{27}$$

## MAP ESTIMATION INSIDE A MULTI-STEP DIFFUSION MODEL

A $T$-step diffusion model defines a sequence of latent variables

$$x_0 \to x_1 \to \cdots \to x_T,$$

generated by the forward noising process

$$q(x_t \mid x_{t-1}) = \mathcal{N}(\alpha_t x_{t-1}, \ (1 - \alpha_t)I).$$

A multi-step reverse sampler learns the reverse Markov chain

$$p_\theta(x_{t-1} \mid x_t) = \mathcal{N}(\mu_\theta(x_t, t), \ \sigma_t^2 I),$$

where the mean is determined using the learned score:

$$\mu_\theta(x_t, t) = \frac{1}{\alpha_t}\left(x_t - \frac{1 - \bar{\alpha}_t}{1 - \alpha_t}\varepsilon_\theta(x_t, t)\right).$$

**Multi-step diffusion is equivalent to MAP with a Gaussian prior**

In the generative model, the reverse conditional satisfies

$$p(x_{t-1} \mid x_t) \propto \exp\left(-\frac{1}{2\sigma_t^2}\|x_{t-1} - \mu_\theta(x_t, t)\|^2\right).$$

For a given observation $Y$ and unknown clean image $X$, classical MAP solves:

$$\hat{X}_{\mathrm{MAP}} = \arg\max_X \left[\log p(Y \mid X) + \log p(X)\right].$$

Identifying $X = x_0$ and $p(X) = p(x_0)$, diffusion models define a hierarchical prior:

$$p(x_0) = \int p(x_T) \prod_{t=1}^{T} p_\theta(x_{t-1} \mid x_t) \, dx_{1:T}.$$

Therefore, the multi-step MAP objective is

$$\hat{x}_0 = \arg\max_{x_0} \left[\log p(Y \mid x_0) + \sum_{t=1}^{T} \log p_\theta(x_{t-1} \mid x_t)\right]. \tag{28}$$

**The score network is the gradient of the log-prior**

From DDPM:
$$\nabla_{x_t} \log p_t(x_t) = -\frac{1}{1 - \bar{\alpha}_t} \, \varepsilon_\theta(x_t, t).$$

Thus diffusion implicitly learns
$$\nabla_{x_0} \log p(x_0),$$
which is the exact prior gradient appearing in MAP.

MAP combines:

$$\nabla_{x_0} \log p(x_0) \quad \text{(diffusion score)} \qquad \text{and} \qquad \nabla_{x_0} \log p(Y \mid x_0) \quad \text{(likelihood)}.$$

**Replacing diffusion's prior with an MRF prior**

We introduce a structured MRF prior:

$$\log p(X) \propto -E_{\mathrm{MRF}}(X).$$

Thus we can replace or augment the diffusion prior term

$$\log p_\theta(x_{t-1} \mid x_t)$$

with

$$-\lambda \, E_{\mathrm{MRF}}(x_t).$$

**Multi-step Reverse With MAP Regularization**

The DDPM update is,
$$x_{t-1} = \mu_\theta(x_t, t) + \sigma_t z.$$

Under MAP regularization, the update becomes:

$$x_{t-1} = \mu_\theta(x_t, t) - \eta_t \nabla_{x_t} E_{\mathrm{MRF}}(x_t) + \sigma_t z. \tag{29}$$

More compactly:

$$x_{t-1} = x_{t-1}^{\mathrm{DDPM/DDIM}} - \eta_t \nabla_{x_t} E_{\mathrm{MRF}}(x_t).$$

**Single-step diffusion as a limiting case**

Our one-step estimator:
$$x_0' = x_T + \sigma_T g_\phi(x_T)$$

is the $T \to 1$ collapse of equation 29.

If

$$\eta = \sum_{t=1}^{T} \eta_t,$$

then the MAP update becomes:

$$x_0 \approx x_T - \eta \nabla_{x_T} E_{\mathrm{MRF}}(x_T).$$

Our learned enhancer satisfies:

$$g_\phi(x_T) \approx -\nabla_{x_T} E_{\mathrm{MRF}}(x_T).$$

Thus:

$$x_0' = x_T + \sigma_T g_\phi(x_T)$$

is a learned proximal/MAP update.

**Final Integrated Statement**

A $T$-step diffusion model defines the prior

$$p(x_0) = \int p(x_T) \prod_{t=1}^{T} p_\theta(x_{t-1} \mid x_t) \, dx_{1:T},$$

and its reverse process

$$p_\theta(x_{t-1} \mid x_t) = \mathcal{N}\big(\mu_\theta(x_t, t), \, \sigma_t^2 I\big)$$

approximates $\nabla_{x_0} \log p(x_0)$.

Thus MAP inference with a likelihood term becomes:

$$\hat{x}_0 = \arg\max_{x_0} \left[ \log p(Y \mid x_0) + \sum_{t=1}^{T} \log p_\theta(x_{t-1} \mid x_t) \right].$$

Substituting the Gaussian form yields the multi-step MAP update:

$$x_{t-1} = \mu_\theta(x_t, t) - \eta_t \nabla_{x_t} E_{\mathrm{MRF}}(x_t) + \sigma_t z.$$

Our single-step model is a limiting case of this regularized reverse diffusion chain.

## ADDITIONAL RESULTS:

We present additional comparative results with existing diffusion model-based methods in Fig. 8. Our method demonstrates superior performance, particularly in recovering fine structures such as artificial flower petals, leaf textures, and cloth patterns, under both ground truth and non-ground truth scenarios. In addition, we provide further quantitative comparisons with state-of-the-art one-step diffusion-based image super-resolution methods, as reported in Table 6.

**Multistep Formulation:** As analyzed in the previous section, MAP regularization only needs to be applied to the intermediate latent states $\{x_t\}$ that participate in the reverse diffusion process. In particular, the MAP-corrected update (Eq. equation 29) adjusts the DDPM/DDIM mean by incorporating the gradient of the MRF energy, thereby refining the predicted sample at each step. To flexibly evaluate the effect of MAP regularization across different diffusion horizons, we apply the correction term at several pre-selected timesteps, corresponding to varying numbers of sampling iterations. Once trained, the starting timestep for the reverse chain can be freely chosen at inference time, providing a controllable trade-off between fidelity (larger number of MAP-corrected steps) and realism (fewer steps, closer to the learned prior), analogous to the behavior observed in classical multi-step diffusion samplers. The total number of steps used during inference is determined jointly by the chosen starting timestep and the skipping stride of the accelerated DDIM-style sampler. For clarity, we report detailed results for 1 to 5 MAP-corrected steps in Table 7, reflecting a practical trade-off between performance and computational budget.

| Datasets | Methods | Evaluation Metrics | | | | | | | | |
|---|---|---|---|---|---|---|---|---|---|---|
| | | PSNR ↑ | LPIPS ↓ | MUSIQ ↑ | CLIPIQA ↑ | SSIM ↑ | DISTS ↓ | FID ↓ | NIQE ↓ | MANIQA ↑ |
| DIV2K-Val | S3Diff | 23.40 | **0.2571** | 68.21 | 0.7007 | 0.5953 | **0.1730** | 19.35 | 4.7391 | 0.4538 |
| | TSD-SR | 23.02 | 0.2673 | **71.69** | **0.7416** | 0.5808 | 0.1821 | 29.16 | **4.3244** | 0.6192 |
| | Add-SR | 23.26 | 0.3623 | 63.39 | 0.5734 | 0.5902 | 0.2123 | 29.68 | 4.7610 | 0.5637 |
| | OSEDiff | 23.72 | 0.2941 | 67.97 | 0.6683 | 0.6108 | 0.1976 | 26.32 | 4.7097 | 0.6148 |
| | SinSR | 24.41 | 0.3240 | 62.82 | 0.6471 | 0.6018 | 0.2066 | 35.57 | 6.0159 | 0.5386 |
| | PassionSR | 24.34 | 0.3440 | 51.19 | 0.4802 | 0.7097 | 0.2075 | 28.45 | 7.039 | 0.2267 |
| | AdcSR | 23.74 | 0.2853 | 68.00 | 0.6764 | 0.6017 | 0.1899 | 25.52 | 4.36 | 0.6090 |
| | StructSR | 23.60 | 0.3286 | 65.02 | 0.6558 | 0.5835 | - | - | - | - |
| | SAM-DiffSR | 23.48 | - | - | - | 0.6042 | - | 25.76 | - | 0.5959 |
| | PiSA-SR | 23.87 | 0.2823 | 69.68 | 0.6927 | 0.6058 | 0.1934 | 25.07 | 4.55 | 0.6400 |
| | TVT | 24.23 | 0.2773 | 68.67 | 0.6986 | 0.6292 | 0.1860 | - | - | - |
| | DMAPSR | **24.58** | 0.2938 | 68.52 | 0.6842 | **0.6112** | 0.1962 | 27.19 | 4.6721 | **0.6416** |
| RealSR | S3Diff | 25.03 | 0.2699 | 67.89 | 0.6722 | 0.7321 | **0.1996** | 108.88 | 5.3311 | 0.4563 |
| | TSD-SR | 24.81 | 0.2743 | **71.19** | **0.7160** | 0.7172 | 0.2104 | 114.45 | **5.1298** | 0.6347 |
| | Add-SR | 24.79 | 0.3091 | 66.18 | 0.5722 | 0.7077 | 0.2191 | 132.05 | 5.5440 | 0.6098 |
| | PassionSR | 22.52 | 0.4913 | 43.21 | 0.3089 | 0.6255 | 0.3185 | 129.54 | 5.706 | 0.2396 |
| | OSEDiff | 25.15 | 0.2921 | 69.09 | 0.6693 | 0.7341 | 0.2128 | 123.49 | 5.6476 | 0.6326 |
| | SinSR | 26.28 | 0.3188 | 60.80 | 0.6122 | 0.7347 | 0.2353 | 135.93 | 6.2872 | 0.5385 |
| | AdcSR | 25.47 | 0.2885 | 69.90 | 0.6731 | 0.7301 | 0.2129 | 118.41 | 5.35 | 0.6360 |
| | StructSR | 25.09 | 0.3610 | 63.93 | 0.6487 | 0.6938 | - | - | - | - |
| | PiSA-SR | 25.50 | 0.2672 | 70.15 | 0.6702 | 0.7417 | 0.2044 | 124.09 | 5.50 | 0.6365 |
| | TVT | 25.81 | **0.2587** | 69.89 | 0.6882 | 0.7396 | 0.2061 | - | - | - |
| | DiT4SR | - | 0.319 | 68.073 | 0.550 | - | - | - | - | 0.661 |
| | InvSR | 24.5 | 0.2872 | 67.4586 | 0.6918 | 0.7262 | - | - | 4.2189 | - |
| | DMAPSR | **26.29** | 0.2918 | 69.81 | 0.6651 | **0.7426** | 0.2178 | 127.92 | 5.4104 | **0.6492** |
| DRealSR | S3Diff | 26.89 | 0.3122 | 64.19 | 0.7122 | 0.7469 | 0.2120 | **119.86** | 6.1647 | 0.4508 |
| | TSD-SR | 27.77 | 0.2967 | **66.62** | **0.7344** | 0.7559 | 0.2136 | 134.98 | **5.9131** | 0.5874 |
| | Add-SR | 27.77 | 0.3196 | 60.85 | 0.6188 | 0.7722 | 0.2242 | 150.18 | 6.9321 | 0.5490 |
| | SinSR | 28.36 | 0.3665 | 55.33 | 0.6383 | 0.7515 | 0.2485 | 170.57 | 6.9907 | 0.4884 |
| | OSEDiff | 27.92 | 0.2968 | 64.65 | 0.6963 | 0.7835 | 0.2165 | 135.30 | 6.4902 | 0.5899 |
| | AdcSR | 28.10 | 0.3046 | 66.26 | 0.7049 | 0.7726 | 0.2200 | 134.05 | 6.45 | 0.5927 |
| | StructSR | 27.98 | 0.3640 | 59.76 | 0.6176 | 0.7566 | - | - | - | - |
| | PiSA-SR | 28.31 | 0.2960 | 66.11 | 0.6970 | 0.7804 | 0.2169 | 130.61 | 6.20 | 0.6156 |
| | TVT | 28.27 | 0.2900 | 65.56 | 0.7220 | 0.7839 | 0.2205 | - | - | - |
| | DiT4SR | - | 0.365 | 64.95 | 0.5480 | - | - | - | - | **0.6270** |
| | DMAPSR | 28.32 | **0.2957** | 64.97 | 0.6975 | **0.7842** | **0.2096** | 139.75 | 6.3245 | 0.6172 |

Table 6: We conduct a quantitative comparison of DMAPSR with state-of-the-art Real-ISR models based on one-step diffusion frameworks across various datasets. The best-performing method is highlighted in bold.

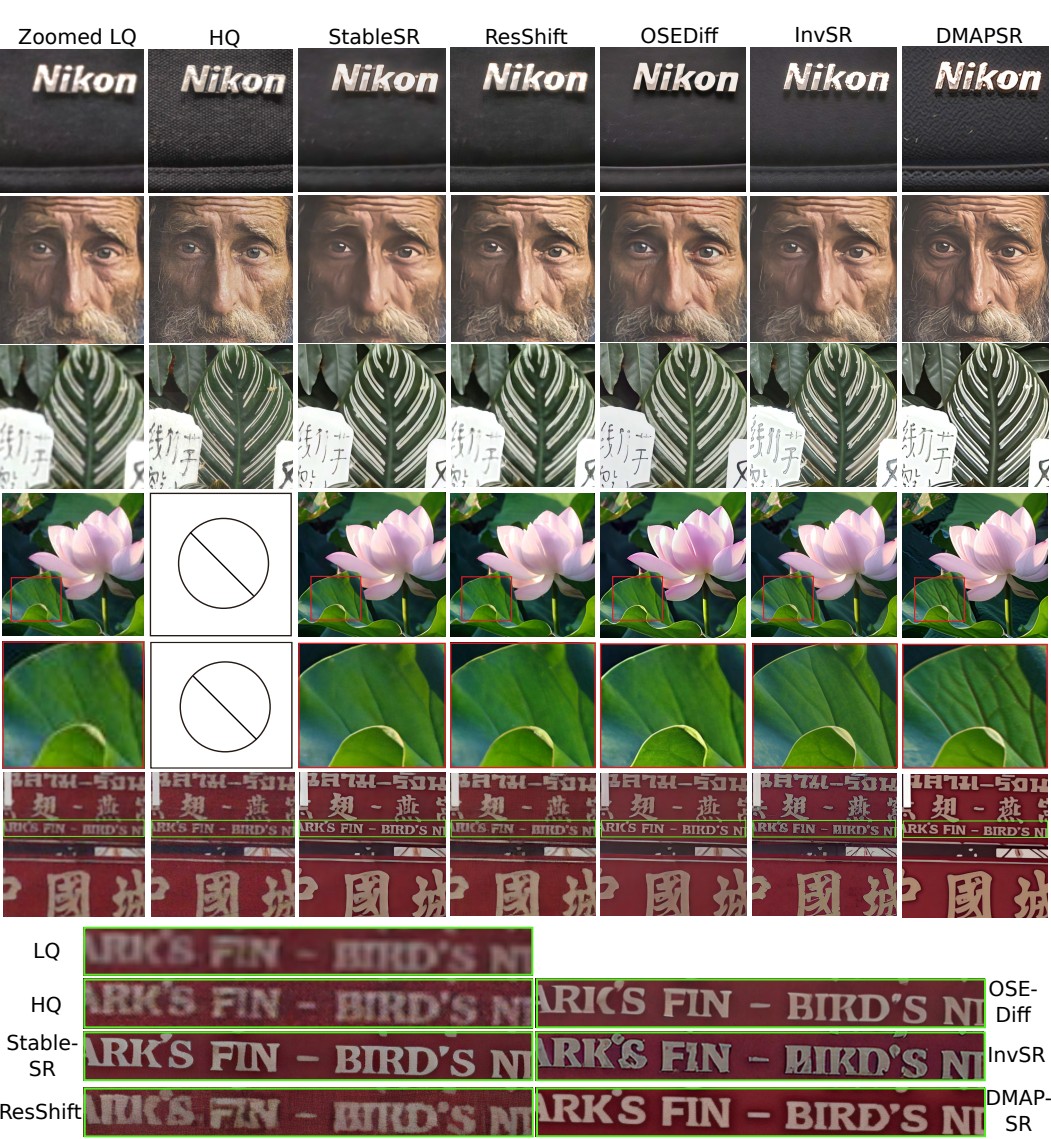

Figure 8: Qualitative visual comparisons of Real-ISR methods are presented. Note that the third example lacks a corresponding high-quality ground truth image. Please zoom in for a clearer view.

Table 7: Ablation on the number of MAP-regularized reverse diffusion steps. Increasing the number of steps improves fidelity by repeatedly applying the MAP-corrected update (Eq. 29), while fewer steps favor realism and efficiency.

| Steps | PSNR ↑ | LPIPS ↓ | MUSIQ ↑ | CLIPIQA ↑ | SSIM ↑ | DISTS ↓ | NIQE ↓ | MANIQA ↑ |
|---|---|---|---|---|---|---|---|---|
| 1 | 28.32 | 0.2957 | 64.97 | 0.6975 | 0.7842 | 0.2096 | 6.3245 | 0.6172 |
| 2 | 28.35 | 0.2954 | 65.02 | 0.6982 | 0.7851 | 0.2087 | 6.3217 | 0.6175 |
| 3 | 28.12 | 0.2972 | 65.14 | 0.6992 | 0.7847 | 0.2092 | 6.3198 | 0.6162 |
| 4 | 28.15 | 0.2961 | 65.21 | 0.6986 | 0.7855 | 0.2099 | 6.3157 | 0.6181 |
| 5 | 28.25 | 0.2941 | 66.12 | 0.7005 | 0.7844 | 0.2093 | 6.3106 | 0.6185 |

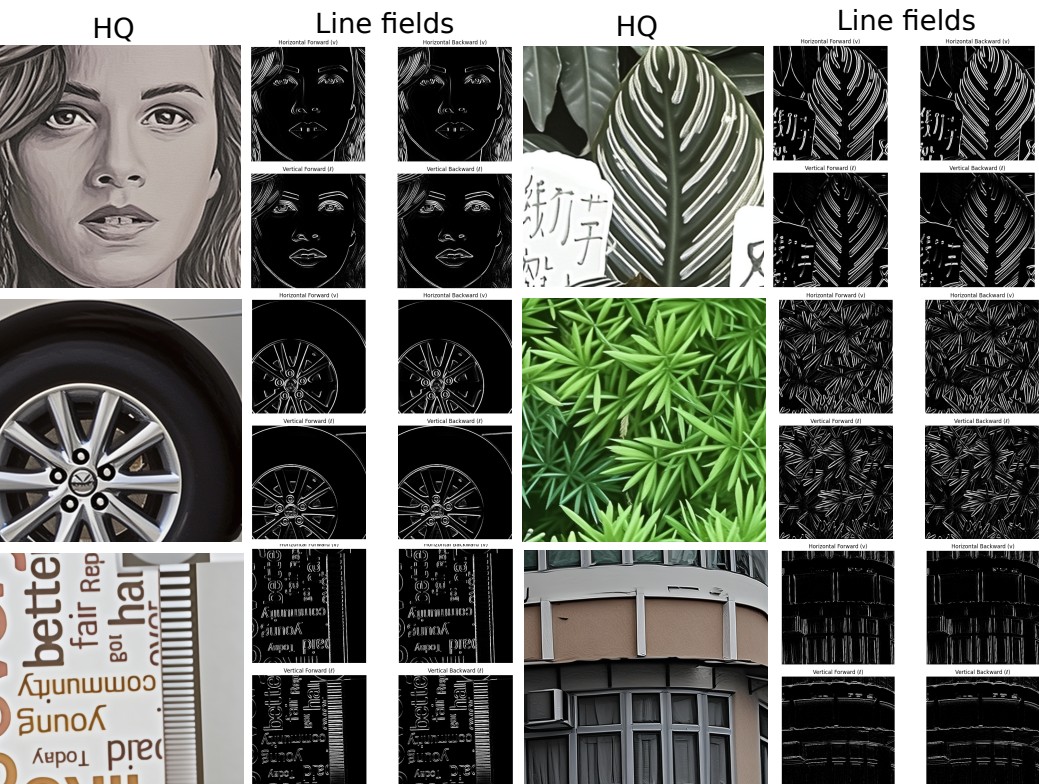

Figure 9: Horizontal and vertical line fields in both forward and backward directions are shown alongside the generated HQ image. Please zoom in for a clearer view.

## VISUALIZATION OF LINE FIELDS:

We present the horizontal and vertical line fields in forward and backward directions generated during inference in Fig. 9.

## USAGE OF LARGE LANGUAGE MODEL:

We have utilized a large language model (LLM) solely for grammatical correction, word choice refinement, and improving sentence phrasing.

