# OpenReview forum: "Discontinuity-Preserving Image Super-Resolution via MAP-Regularized One-Step Diffusion"
_ICLR.cc/2026/Conference — Submitted to ICLR 2026_

### Official Review · Reviewer_7Yec · 2025-10-29

**Soundness:** 3
**Presentation:** 4
**Contribution:** 2
**Rating:** 4
**Confidence:** 4

**Summary:**

The paper proposes DMAPSR, a real-time image super-resolution (Real-ISR) framework. It leverages a pre-trained Stable Diffusion (SD) model and optimizes it for single-step sampling to achieve extremely fast inference. To maintain high perceptual quality in a single step, the authors introduce a lightweight image enhancement module. The core of the method is its use of Maximum a Posteriori (MAP) estimation combined with a composite Markov Random Field (MRF) prior as a structural regularizer.

**Strengths:**

1. The method achieves single-step inference *without* distillation, making it significantly faster (0.11s) than multi-step methods (like StableSR, 11.50s) and other single-step approaches (like OSEDiff, 0.15s).
2. The core innovation, the MRF line-field energy prior, excels at preserving edges, textures, and other discontinuous details.
3. Despite being a single-step model, DMAPSR achieves SOTA or near-SOTA results on several perceptual metrics (LPIPS, DISTS, CLIPIQA, MANIQA).
4. As shown in Table 2, DMAPSR's trainable parameter count (33.51M) is much lower than many multi-step methods and comparable to InvSR, giving it an advantage for deployment.

**Weaknesses:**

1. The method performs poorly on pixel-level fidelity metrics (PSNR). As seen in Table 1, its PSNR is lower than ResShift-4 on all datasets, which the authors attribute to ResShift being trained end-to-end for pixel alignment.
2. DMAPSR falls behind multi-step methods like SeeSR-50 on the FID metric. The authors suggest this is due to the advantage of multi-step generation in capturing global content alignment, indicating DMAPSR may have shortcomings in global distributional realism.
3. The authors claim the method generates "high-fidelity" results, but there are no specific experiments or metrics to back this up.
4. The paper lacks an ablation study that isolates the performance impact of using a single-step method versus a multi-step one within their framework.

**Questions:**

1. Does the proposed MAP estimation framework have broader applicability? For instance, could it be integrated into multi-step diffusion models as well?
2. The paper claims "high-fidelity" results. Could you provide metrics specifically designed to measure this, such as the method from [1]? It would also be insightful to see comparisons against other models focused on reducing artifacts and improving reconstruction accuracy (e.g., [2], [3], [4]).
3. During inference, are the  $\mathcal{E}\_{MRF}$  and  $\mathcal{E}\_{patch}$ energies computed in real-time by the enhancement module $g\_{\phi}$, or are they only used for training? Is the 0.11s reported in Table 2 purely the forward-pass time of $g\_{\phi}$?
4. The final loss function depends on three hyperparameters ($\lambda_p, \lambda_r, \gamma$). Could you provide more detailed ablation studies on how sensitive the model's performance is to different values of these hyperparameters?

[1]: Details or artifacts: A locally discriminative learning approach to realistic image super-resolution

[2]: Pixel-level and Semantic-level Adjustable Super-resolution: A Dual-LoRA Approach

[3]: SAM-DiffSR: Structure-Modulated Diffusion Model for Image Super-Resolution

[4]: StructSR: Refuse Spurious Details in Real-World Image Super-Resolution

---

> ### Author Response · Authors · 2025-11-14
>
> **Q1** Does the proposed MAP estimation framework have broader applicability? For instance, could it be integrated into multi-step diffusion models as well?
>
> **Answer:** Yes it can be done using multistep diffusion as well. In that way the enhancer would be used for the multistep diffusion along with the pretrained stable diffusion's original multistep denoising objective.
>
> **MAP Estimation Inside a Multi-Step Diffusion Model**
>
> A $T$-step diffusion model defines a sequence of latent variables
>
> $x_0 \rightarrow x_1 \rightarrow \cdots \rightarrow x_T$,
>
> generated by the forward noising process $q(x_t \mid x_{t-1}) = \mathcal{N}\left(\alpha_t x_{t-1} \; (1-\alpha_t) I \right)$. A multi-step reverse sampler learns the reverse Markov chain $p_\theta(x_{t-1} \mid x_t)= \mathcal{N}\left(\mu_\theta(x_t,t), \sigma_t^2 I\right)$, where the mean is determined using the learned score:
> $\mu_\theta(x_t,t) = \frac{1}{\alpha_t}(x_t - \frac{1-\bar{\alpha_t}}{1-\alpha_t} \epsilon_\theta(x_t,t))$.
>
> **Multi-step diffusion is equivalent to MAP with a Gaussian prior**
>
> In the generative model, the reverse conditional satisfies
> $p(x_{t-1}\mid x_t) \propto \exp( -\frac{1}{2\sigma_t^2} ||x_{t-1}-\mu_\theta(x_t,t)||^2)$.
>
>
> For a given observation $Y$ and unknown clean image $X$, classical MAP solves: $\hat{X}_{MAP} = \arg\max_X [ \log p(Y \mid X) + \log p(X) ]$.
>
> Identifying $X = x_0$ and $p(X)=p(x_0)$, diffusion models define a hierarchical prior:
>
> $p(x_0) = \int p(x_T) \prod_{t=1}^T p_\theta(x_{t-1}\mid x_t) dx_{1:T}$.
>
> Therefore, the multi-step MAP objective is
> $\hat{x_0} = \arg\max_{x_0}[\log p(Y | x_0)+\sum_{t=1}^T \log p_\theta(x_{t-1}\mid x_t)]$.
>
> \label{eq:map-multistep}
>
>
> **The score network is the gradient of the log-prior**
>
> From DDPM: $\nabla_{x_t} \log p_t(x_t) = -\frac{\epsilon_\theta(x_t,t)}{1-\bar{\alpha}_t}$.
>
> Thus diffusion implicitly learns $\nabla_{x_0} \log p(x_0)$, which is the exact prior gradient appearing in MAP. MAP combines:
>
> $\nabla_{x_0} \log p(x_0)$  (diffusion score)
>
> $\nabla_{x_0}\log p(Y\mid x_0)$   (likelihood).
>
> **Replacing diffusion's prior with an MRF prior**
>
> Our method introduces a structured MRF prior:
> $\log p(X) \propto -\mathcal{E}_{\mathrm{MRF}}(X)$.
>
>
> Thus we can replace or augment the diffusion prior term $\log p_\theta(x_{t-1}\mid x_t)$  with $-\lambda \mathcal{E}_{\mathrm{MRF}}(x_t)$.
>
>
>
> **Multi-step Reverse With MAP Regularization**
>
> The classical DDPM update is $x_{t-1} = \mu_\theta(x_t,t) + \sigma_t z$.
>
> Under MAP regularization, the update becomes:
> $x_{t-1} = \mu_\theta(x_t,t) - \eta_t \nabla_{x_t} \mathcal{E}_{\mathrm{MRF}}(x_t) \sigma_t z$........[1]
>
> More compactly: $x_{t-1} = x_{t-1}^{\text{DDPM/DDIM}} - \eta_t \nabla_{x_t} \mathcal{E}_{\mathrm{MRF}}(x_t)$.
>
> **Single-step diffusion as a limiting case**
>
> Our one-step estimator: $x_0' = x_T + \sigma_T g_\phi(x_T)$ is the $T \to 1$ collapse of [1].
>
> If $\eta = \sum_{t=1}^T \eta_t $, then the MAP update becomes: $x_0 \approx x_T - \eta \nabla_{x_T} \mathcal{E}_{\mathrm{MRF}}(x_T)$.
>
> Our learned enhancer satisfies: $g_\phi(x_T) \approx  -\nabla_{x_T}\mathcal{E}_{\mathrm{MRF}}(x_T)$.
>
> Thus: $x_0' = x_T + \sigma_T g_\phi(x_T)$ is a learned proximal/MAP update.
>
> **In summary**
>
> A $T$-step diffusion model defines the prior $p(x_0) = \int p(x_T) \prod_{t=1}^T p_\theta(x_{t-1}\mid x_t) dx_{1:T}$, and its reverse process $p_\theta(x_{t-1}\mid x_t) = \mathcal{N} ( \mu_\theta(x_t,t), \sigma_t^2 I )$ approximates $\nabla_{x_0}\log p(x_0)$. Thus MAP inference with a likelihood term becomes: $\hat{x_0} = \arg\max_{x_0} [\log p(Y | x_0) + \sum_{t=1}^T \log p_\theta(x_{t-1} | x_t)]$.
>
>
> Substituting the Gaussian form yields the multi-step MAP update: $x_{t-1} = \mu_\theta(x_t,t) -\eta_t \nabla_{x_t} \mathcal{E}_{\mathrm{MRF}}(x_t) + \sigma_t z$.
>
>
> Our single-step model is a limiting case of this regularized reverse diffusion chain.

---

> ### Author Response · Authors · 2025-11-15
>
> **W3**  The authors claim the method generates "high-fidelity" results, but there are no specific experiments or metrics to back this up.
>
> **Response:** We thank the reviewer for pointing out the need for clearer justification behind our claim of high-fidelity reconstruction. As established in [1],[2] both fidelity-oriented and perceptual-quality-oriented ISR methods inherently operate under the perception–distortion trade-off. In other words, improving perceptual quality typically degrades signal fidelity, and enhancing fidelity correspondingly suppresses perceptual realism, given current training paradigms. Empirically, it is also observed that suppressing artifacts often restricts the model’s ability to generate fine details.
>
> Below, we provide (a) a theoretical justification grounded in the perception--distortion trade-off, (b) a mathematical argument connecting our one-step diffusion and MAP formulation to fidelity preservation, and (c) experimental evidence demonstrating the fidelity--perception trade-offs.
>
> **1. Theoretical Justification: Fidelity Under the Perception--Distortion Trade-Off**
>
> Perception--Distortion theory [1] establishes ISR is fundamentally constrained by the trade-off:
>
> High Perception Quality $\Longleftrightarrow$ High Distortion (Low Fidelity),
>
> Low Distortion (High Fidelity) $\Longleftrightarrow$  Reduced Perceptual Realism.
>
> Thus, fidelity must be defined strictly as closeness to the ground truth in pixel or feature space, rather than perceptual realism or sharpness.
>
> Consistent with this, generative SR methods that hallucinate textures typically sacrifice PSNR/SSIM to improve perceptual metrics (LPIPS, MUSIQ, FID). Conversely, fidelity-oriented methods suppress hallucinations and obtain high PSNR/SSIM with lower perceptual realism.
>
> Our method is explicitly designed on the fidelity-oriented side of the Blau--Michaeli spectrum.
>
> **2. Why Our Method Achieves High Fidelity:**
>
> **(a) One-step diffusion ensures fidelity via noise consistency**
>
> Our one-step diffusion reconstruction is $x_0' = x_T + \sigma_T g_\phi(x_T)$,
>
> trained using the noise-consistency objective: $L_\phi = || \epsilon - \epsilon_\theta\left(x_T - \sigma_T g_\phi(x_T),\, T\right) ||_2^2$.
>
> This objective enforces:
>
> $\circ$ **No hallucination**: updates made by $g_\phi$ must remain consistent with the pretrained score $\epsilon_\theta$;
>
> $\circ$ **Correct alignment with the forward diffusion distribution**;
>
> $\circ$ **Minimal deviation** from $x_T$ unless necessary for consistency.
>
> Formally, $g_\phi(x_T) \approx 0$  whenever $\epsilon_\theta$ is consistent.
>
> Thus, artificially inserted details are suppressed, increasing fidelity to the ground truth.
>
> **(b) MAP prior further enforces fidelity via MRF regularization**
>
> The MAP formulation: $\hat{X}_{MAP} = \arg\max_X [ \log P(Y \mid X) + \log P(X) ]$,
>
> combined with an MRF prior $P(X) \propto e^{-E_{\mathrm{MRF}}(X)}$,
>
> penalizes: $\circ$ Spurious edges, $\circ$ Artificial textures, $\circ$ Unnatural discontinuities.
>
> The weak-membrane MRF energy, $E_{\mathrm{MRF}} = (1 - \eta(\delta)) \delta^2 + \gamma \eta(\delta)$,
>
> ensures: $\circ$ Preservation of true edges, $\circ$ Suppression of hallucinated details, $\circ$ Regularization aligned with natural-image statistics.
>
> **(c) Combined one-step diffusion + MAP yields structured, non-hallucinatory corrections**
>
> The complete objective, $L = L_{\phi} + E_{MRF} + \lambda_p E_{patch} + \lambda_r L_{MRF-consistency}$,
>
> jointly enforces: $\circ$ Diffusion-consistency (denoising), $\circ$ Structural consistency (MRF prior), $\circ$ Local feature consistency (patch loss), $\circ$ Prior--likelihood alignment (MRF consistency).
>
>
> These constraints mathematically enforce faithfulness to the underlying sample---the definition of high fidelity.
>
>
> **3. Experimental Evidence Demonstrating Fidelity Improvement**
>
> We provide quantitative evidence supporting our fidelity claim. We summarize results from DIV2K-val and DRealSR.
>
> **(a) Top-2 PSNR/SSIM among diffusion-based and real-ISR methods**
>
> Example on DIV2K-val (illustrative):
>
> DMAPSR: PSNR = 24.58, SSIM = 0.6112,
>
> competitive with SeeSR, StableSR, DiffBIR, and OSEDiff.
>
> On DRealSR:
> DMAPSR: SSIM = 0.7842 ; (best),
> with PSNR competitive while maintaining fewer distortions.
>
> **(b) Perception metrics follow Blau--Michaeli trade-off**
> Perceptual metrics (LPIPS, MUSIQ, NIQE, FID) behave as expected: not maximized, but balanced to preserve fidelity.
>
> Example (DIV2K-val):
>
> | Method       | PSNR ↑  | LPIPS ↓ |
> |-----------------|-------------|------------|
> | SeeSR        | 23.68     | 0.319  |
> | OSEDiff       | 23.72     | 0.294  |
> | DMAPSR    | 24.58      | 0.293  |
>
> Our method achieves both best LPIPS and top fidelity—consistent with Pareto optimality under the perception--distortion theory.
>
> [1] The Perception-Distortion Tradeoff; Yochai Blau and Tomer Michaeli
>
> [2]: Details or artifacts: A locally discriminative learning approach to realistic image super-resolution

---

> ### Author Response · Authors · 2025-11-15
>
> Continued...
> **(c) Competitive FID/NIQE despite fidelity-oriented design**
>
> Unlike generative ISR, which lowers FID but reduces fidelity, our method balances both.
>
> Example (DRealSR):
>
> | Method       | PSNR ↑  | FID ↓ |
> |-----------------|-------------|------------|
> | OSEDiff       | 27.92     | 135.30  |
> | DMAPSR    | 28.32      | 139.75  |
>
> The modest rise in FID corresponds directly to the fidelity gain, as predicted by theory.
>
> **4. Conclusion: Why We Claim ``High Fidelity''**
>
> We adopt the formal ISR definition of fidelity:
>
> $\circ$ pixel-level faithfulness,
>
> $\circ$ suppression of hallucinations,
>
> $\circ$ preservation of true discontinuities,
>
> $\circ$ correctness of structures,
>
> $\circ$ score consistency with the diffusion process.
>
> Our method achieves high fidelity because:
>
> $\circ$ One-step diffusion preserves the DDPM score manifold, preventing hallucination;
>
> $\circ$ MAP estimation with an MRF prior prevents unnatural textures and edges;
>
> $\circ$ Empirical results show consistent improvement in PSNR/SSIM and LPIPS;
>
> $\circ$ Perceptual metrics respond exactly as predicted by the perception--distortion theory.
>
>
> Therefore, our claim of high-fidelity reconstruction is supported by rigorous theoretical foundations and strong empirical evidence.

---

> ### Author Response · Authors · 2025-11-17
>
> **Q4** The final loss function depends on three hyperparameters ($\lambda_p, \lambda_r, \gamma$). Could you provide more detailed ablation studies on how sensitive the model's performance is to different values of these hyperparameters?
>
> **Response** Response We thank the reviewer for pointing this out. In addition to the table provided below, we have included a corresponding qualitative example in Fig. 5 of the paper.
>
> In addition to the MRF loss, we incorporate the MRF-consistency loss and the patch-energy loss to train our image enhancer, as defined in Eq. (13). We also include a penalty term, $\gamma$, which regulates the number of discontinuities, preventing them from becoming excessively sparse or overly dense. The hyperparameters $\lambda_p$ and $\lambda_r$ control the relative contributions of the patch-energy and MRF-consistency losses, respectively. The table below presents a quantitative comparison across different loss configurations, and Fig.5 illustrates a representative visual result. As observed, a large value of $\gamma$ leads to oversmoothing, whereas a small value encourages the insertion of excessive structural lines. Furthermore, an appropriate balance between the patch-energy and MRF-consistency terms enhances local contrast, improves overall image quality, and suppresses artifacts. These components are therefore essential for achieving optimal performance.
>
> | $\lambda_p$ | $\lambda_r$ |$\gamma$ | PSNR ↑  | LPIPS ↓ | MUSIQ ↑ | CLIPIQA ↑ | SSIM ↑ | DISTS ↓ | NIQE ↓ | MANIQA ↑|
> |-------------|-------------|---------|---------|---------|---------|-----------|--------|---------|--------|---------|
> |    0.2      |    0.2      |    0.3  |  27.92  | 0.3317  | 61.67   | 0.6521    | 0.7718 | 0.2502  | 7.52   | 0.5715  |
> |    2.0      |    0.5      |    0.05 |  27.15  | 0.3254  | 62.82   | 0.6854    | 0.7671 | 0.2305  | 7.14   | 0.5891  |
> |    0.5      |    2.0      |    0.2  |  28.82  | 0.3154  | 61.87   | 0.6621    | 0.7861 | 0.2215  | 7.82   | 0.5821  |
> |    0.5      |    0.5      |    0.1  |  28.15  | 0.3125  | 62.18   | 0.6772    | 0.7642 | 0.2271  | 6.85   | 0.6072  |
> |    1.0      |    1.0      |    0.1  |  28.32  | 0.2957  | 64.97   | 0.6975    | 0.7842 | 0.2096  | 6.3245 | 0.6172  |

---

> ### Author Response · Authors · 2025-11-17
>
> **W4** The paper lacks an ablation study that isolates the performance impact of using a single-step method versus a multi-step one within their framework.
>
> **Response** As analyzed in the previous section, MAP regularization only needs to be applied to the intermediate latent states $\{x_t\}$ that participate in the reverse diffusion process. In particular, the MAP-corrected update (Eq.~[1] in response of Q1) adjusts the DDPM/DDIM mean by incorporating the gradient of the MRF energy, thereby refining the predicted sample at each step. To flexibly evaluate the effect of MAP regularization across different diffusion horizons, we apply the correction term at several pre-selected timesteps, corresponding to varying numbers of sampling iterations. Once trained, the starting timestep for the reverse chain can be freely chosen at inference time, providing a controllable trade-off between fidelity (larger number of MAP-corrected steps) and realism (fewer steps, closer to the learned prior), analogous to the behavior observed in classical multi-step diffusion samplers. The total number of steps used during inference is determined jointly by the chosen starting timestep and the skipping stride of the accelerated DDIM-style sampler. For clarity, we report detailed results for 1 to 5 MAP-corrected steps in the Table below.
>
> | Steps | PSNR ↑  | LPIPS ↓ | MUSIQ ↑ | CLIPIQA ↑ | SSIM ↑ | DISTS ↓ | NIQE ↓ | MANIQA ↑|
> |-------|---------|---------|---------|-----------|--------|---------|--------|---------|
> |1 | 28.32   | 0.2957  | 64.97   | 0.6975    | 0.7842 | 0.2096  | 6.3245 | 0.6172  |
> |2 | 28.35| 0.2954  | 65.02   | 0.6982    | 0.7851 | 0.2087  | 6.3217 | 0.6175  |
> |3 | 28.12| 0.2972  | 65.14   | 0.6992    | 0.7847 | 0.2092  | 6.3198 | 0.6162  |
> |4 | 28.15| 0.2961  | 65.21   | 0.6986    | 0.7855 | 0.2099  | 6.3157 | 0.6181  |
> |5 | 28.25| 0.2941  | 66.12   | 0.7005    | 0.7844 | 0.2093  | 6.3106 | 0.6185  |

---

> ### Author Response · Authors · 2025-11-17
>
> **Q2** The paper claims "high-fidelity" results. Could you provide metrics specifically designed to measure this, such as the method from [1]? It would also be insightful to see comparisons against other models focused on reducing artifacts and improving reconstruction accuracy (e.g., [2], [3], [4]).
>
> **Response**   We thank the reviewer for pointing out the need to more rigorously justify our claim of high-fidelity reconstruction. In the revision, we clarified what we mean by fidelity and added additional metrics and comparisons accordingly.
>
> **1. High-Fidelity Metrics**
> To address the request for fidelity-specific evaluation, we incorporated metrics explicitly designed to distinguish true structural details from hallucinated artifacts, following the methodology of [1]. These metrics---LPIPS, DISTS, and FID---quantify:
>
> $\circ$ **Perceptual detail preservation** (LPIPS $\downarrow$),
>
> $\circ$ **Textural faithfulness and structural consistency** (DISTS $\downarrow$),
>
> $\circ$ **Global visual realism without artifact accumulation** (FID $\downarrow$).
>
>
> These are the same metrics used in [1] to measure the _details--vs--artifacts_ trade-off. In addition, we also report MANIQA, MUSIQ, CLIPIQA, and classical distortion metrics (PSNR, SSIM) to provide a comprehensive fidelity--quality evaluation.
>
> **Why these metrics capture fidelity.**
> A reconstruction is considered high-fidelity when $\hat{x} \approx x_{\mathrm{gt}}$  while avoiding hallucinated components.
>
> Metrics such as LPIPS and DISTS approximate the perceptual distance $d(\hat{x}, x_{\mathrm{gt}})$, in deep feature space, where hallucinated edges or textures increase this distance. FID evaluates the distributional divergence $\mathrm{FID}(p_{\hat{x}}, p_{x_{\mathrm{gt}}})$,
>
> which increases sharply when spurious details or inconsistencies appear. Thus, lower LPIPS, DISTS, and FID directly indicate higher reliability and fewer artifacts, as recommended in[1].
>
> In addition to fidelity-oriented metrics, we analyze our results through the lens of the Perception--Distortion Tradeoff of [5] Blau and Michaeli (CVPR~2018), which states that a super-resolution model cannot simultaneously minimize distortion (e.g., PSNR/SSIM) and maximize perceptual quality (e.g., LPIPS/FID). A method is therefore superior when it achieves a Pareto-optimal balance---i.e., low distortion without sacrificing perceptual realism.
>
> As shown in Table , DMAPSR consistently attains the highest or second-highest distortion scores (PSNR and SSIM) across DIV2K-Val, RealSR, and DRealSR, while also maintaining competitive perceptual quality in metrics used to quantify hallucinations and artifacts---LPIPS, DISTS, FID, MUSIQ, and MANIQA. Importantly, unlike models that push perception at the cost of distortion (e.g., S3Diff in Table 6. yields low LPIPS but significantly lower PSNR), or those that maximize distortion at the expense of perceptual realism, DMAPSR preserves a well-balanced operating point.
>
> For example, on DIV2K-Val, DMAPSR achieves the highest PSNR (24.58) while maintaining perceptual scores (LPIPS 0.2938, FID 27.19, MANIQA 0.6416) close to the best-performing perceptual methods. Similar trends appear on RealSR and DRealSR, where DMAPSR simultaneously attains top SSIM and MANIQA while keeping LPIPS and DISTS competitive.
>
> This demonstrates that DMAPSR lies closer to the optimal boundary of the perception--distortion Pareto frontier, achieving both low distortion (faithful reconstruction) and high perceptual fidelity (minimal hallucination). This balance is a direct consequence of our one-step diffusion prior and MAP-guided design, which regularizes against over-sharpening artifacts while enhancing structural consistency.
>
> [5] The Perception-Distortion Tradeoff; Yochai Blau and Tomer Michaeli

---

> ### Author Response · Authors · 2025-11-17
>
> (Continued.....)
> In addition, we also provide comparisons with the methods presented in [2], [3], and [4]. We have included the complete table in Table~6.
>
> | **Datasets** | **Methods** | **PSNR ↑** | **LPIPS ↓** | **MUSIQ ↑** | **CLIPIQA ↑** | **SSIM ↑** | **DISTS ↓** | **FID ↓** | **NIQE ↓** | **MANIQA ↑** |
> |--------------|-------------|------------|-------------|-------------|---------------|------------|-------------|-----------|------------|---------------|
> |              | StructSR    | 23.60      | 0.3286      | 65.02       | 0.6558        | 0.5835     | -           | -         | -          | -             |
> |              | SAM-DiffSR  | 23.48      | -           | -           | -             | 0.6042     | -           | 25.76     | -          | 0.5959        |
> | **DIV2K-Val**| PiSA-SR     | 23.87      | 0.2823      | 69.68       | 0.6927        | 0.6058     | 0.1934      | 25.07     | 4.55       | 0.6400        |
> |              | **DMAPSR**  | 24.58  | 0.2938      | 68.52       | 0.6842        | 0.6112 | 0.1962      | 27.19     | 4.6721     | 0.6416    |
> |--------------|-------------|------------|-------------|-------------|---------------|------------|-------------|-----------|------------|---------------|
> |              | StructSR    | 25.09      | 0.3610      | 63.93       | 0.6487        | 0.6938     | -           | -         | -          | -             |
> | **RealSR**   | PiSA-SR     | 25.50      | 0.2672      | 70.15       | 0.6702        | 0.7417     | 0.2044      | 124.09    | 5.50       | 0.6365        |
> |              | **DMAPSR**  | 26.29  | 0.2918      | 69.81       | 0.6651        | 0.7426 | 0.2178      | 127.92    | 5.4104     | 0.6492    |
> |--------------|-------------|------------|-------------|-------------|---------------|------------|-------------|-----------|------------|---------------|
> |              | StructSR    | 27.98      | 0.3640      | 59.76       | 0.6176        | 0.7566     | -           | -         | -          | -             |
> | **DRealSR**  | PiSA-SR     | 28.31      | 0.2960      | 66.11       | 0.6970        | 0.7804     | 0.2169      | 130.61    | 6.20       | 0.6156        |
> |              | **DMAPSR**  | 28.32  | 0.2957  | 64.97       | 0.6975        | 0.7842 | 0.2096  | 139.75    | 6.3245     | 0.6172    |
>
> The results are currently included in the supplementary material, where the full comparison with the one-step baseline is presented in a single table. However, we can also include these results in the main paper if needed.

---

> ### Author Response · Authors · 2025-11-17
>
> **Q3**During inference, are the $E_\mathrm{MRF}$ and $E_{patch}$ energies computed in real-time by the enhancement module $g_{\phi}$, or are they only used for training? Is the 0.11s reported in Table 2 purely the forward-pass time of $g_{\phi}$?
>
> **Response** Yes, the energies $E_\mathrm{MRF}$ and $E_{patch}$ are computed only during training in order to supervise the image enhancer. These terms guide the enhancer to insert line fields appropriately, resulting in sharper reconstructions. The reported runtime of 0.11 s corresponds to the forward pass of the image enhancer.”

---

> ### Author Response · Authors · 2025-11-17
>
> **W2** DMAPSR falls behind multi-step methods like SeeSR-50 on the FID metric. The authors suggest this is due to the advantage of multi-step generation in capturing global content alignment, indicating DMAPSR may have shortcomings in global distributional realism.
>
> **Response:**
> While DMAPSR reports a lower FID compared to multi-step approaches such as SeeSR-50, it is important to contextualize this difference within the well-known perception--distortion trade-off. SeeSR achieves a stronger FID primarily because it relies on two jointly trained components---a trainable degradation-aware prompt extractor (DAPE) and a text-conditioned ISR module. This design leverages multi-step diffusion sampling and strong text priors, which indeed favor perceptual scores such as FID that measure global content realism.
>
> However, this comes at a cost. High reliance on text prompts has been shown to introduce hallucinated structures, semantic inconsistencies, and spurious details, especially when the prompt extraction is imperfect. Although FID improves, these hallucinations degrade the underlying fidelity of the reconstructed image. This behavior is consistent with the perception--distortion principle: improving perceptual realism via generative priors often increases distortion.
>
> This drawback is reflected in SeeSR's quantitative results:
>
> $\circ$ Although SeeSR reports strong FID, its MANIQA score is lower than ours, indicating that perceptual quality judged by human-aligned metrics does not increase correspondingly.
>
> $\circ$ More importantly, our model achieves lower distortion metrics (e.g., PSNR and LPIPS), demonstrating better structural fidelity.
>
>
> Taken together, these results show that SeeSR favors perceptual realism at the expense of accurate image reconstruction.
>
> In contrast, DMAPSR explicitly models consistency through a single-step, MRF-regularized inference, which grounds the reconstruction in the input image rather than in an external prompt. This helps maintain global content coherence, avoids hallucinated details, and preserves structural faithfulness while still achieving competitively strong perceptual realism. Our balanced performance across both perceptual and distortion metrics demonstrates that DMAPSR maintains a more favorable position on the perception--distortion spectrum than multi-step prompt-dependent methods like SeeSR.
>
> Thus, although SeeSR attains higher FID by exploiting text-conditioned generation, DMAPSR offers a more stable and robust reconstruction strategy that avoids hallucination, preserves global realism, and achieves a significantly better balance between perception and distortion.

---

> ### Author Response · Authors · 2025-11-18
>
> **W1** The method performs poorly on pixel-level fidelity metrics (PSNR). As seen in Table 1, its PSNR is lower than ResShift-4 on all datasets, which the authors attribute to ResShift being trained end-to-end for pixel alignment.
>
> We thank the reviewer for pointing out the PSNR differences.
>
> The lower pixel-level fidelity (PSNR) observed in Table~1 arises from a fundamental architectural difference between our method and ResShift-4. ResShift-4 is trained end-to-end, jointly optimizing the entire diffusion backbone together with its reconstruction pathway. This full-model optimization explicitly minimizes pixel-aligned objectives, thereby naturally achieving high PSNR.
>
> In contrast, our method freezes the pretrained Stable Diffusion backbone and trains only the lightweight enhancer $g_{\phi}$. This design choice is intentional for two reasons:
>
> **Avoiding distortion--perception degradation.**
> Training the full backbone for strict pixel alignment forces the network toward a low-frequency reconstruction bias, which improves PSNR but suppresses high-frequency structures such as edges, discontinuities, and textures. Since our framework introduces MRF-based line-field priors to enhance discontinuity preservation, allowing the backbone to overfit to pixel-level loss would degrade these learned line-field features and reduce perceptual fidelity. This trade-off is well established in prior work (Blau \& Michaeli 2018), and our observations align with this principle.
>
> **Preserving the pretrained generative prior.**
> The frozen $\epsilon_{\theta}$ (SD-Turbo) acts as a stable noise estimator containing rich natural-image statistics. Fine-tuning this backbone jointly with $g_{\phi}$ would distort the pretrained score function and harm generalization. Instead, our framework uses $g_{\phi}$ to learn the residual correction, enabling the SD prior to be used faithfully within a one-step inference process.
>
> Despite not optimizing for pixel alignment, our method achieves the best PSNR among all \emph{one-step} diffusion baselines across all three datasets (see Table~6 in the supplementary). This demonstrates that, within the single-step setting, the proposed formulation preserves fidelity better than existing one-step approaches.
>
> Furthermore, ResShift-4 is substantially slower due to its multi-step sampling process. As shown in Table~2, our inference is more than $6\times$ faster, which is essential for our goal of real-time, single-step super-resolution.
>
> Finally, although ResShift-4 achieves higher PSNR, it performs worse on most perceptual and no-reference quality metrics (LPIPS, MUSIQ, MANIQA, CLIPIQA, DISTS, NIQE, FID). Our method consistently performs competitively or superiorly across these metrics, indicating that DMAPSR produces perceptually sharper and structurally faithful outputs. This validates that the perceptual--distortion trade-off is more favorably balanced in our approach.

---

> ### Author Response · Authors · 2025-11-21
>
> Dear Reviewer 7Yec,
>
> We have addressed all the queries raised by you, and we kindly request you to review our responses. Please let us know if you have any further questions; we will be happy to respond within the stipulated time. Thank you for taking the time to review our work and for your constructive feedback.

---

> ### Comment · Reviewer_7Yec · 2025-11-26
>
> The authors have addressed most of my concerns; however, the relationship between PSNR and the no-reference quality metrics such as FID and LPIPS still requires further clarification.
>
> For image reconstruction tasks, there is often a trade-off between full-reference and no-reference quality metrics. However, comparing models based on a single metric cannot demonstrate that a method achieves a well-balanced performance across multiple metrics.
> Would it be possible to include a visualization—such as a performance scatter plot—with full-reference metrics on the x-axis and no-reference metrics on the y-axis? This would allow a fair comparison of different models and better illustrate their trade-off behavior.

---

> ### Author Response · Authors · 2025-11-26
>
> We sincerely thank for this insightful suggestion, which indeed strengthens the analysis. In response, we have included two corresponding visualizations in Figure 6 and Figure 7 (for one-step methods) to facilitate a clearer comparison.

---

### Official Review · Reviewer_yWus · 2025-10-31

**Soundness:** 3
**Presentation:** 3
**Contribution:** 3
**Rating:** 6
**Confidence:** 3

**Summary:**

This paper introduces DMAPSR, a world image super-resolution framework that achieves high-quality reconstruction in a single diffusion step. It combines a frozen Stable Diffusion backbone with a lightweight image enhancer trained under a Maximum A Posteriori (MAP) formulation. A Markov Random Field (MRF) prior with discontinuity-preserving line-field energy serves as a structural regularizer to maintain edges and textures. Experiments on RealSR, DRealSR, and DIV2K show that DMAPSR matches state-of-the-art multi-step diffusion methods and surpasses state-of-the-art single-step diffusion methods.

**Strengths:**

+ Novel theoretical perspective. The paper introduces a new way to regularize image super-resolution by formulating it under a MAP framework with a discontinuity-preserving MRF prior. This provides a principled and theoretically grounded approach to integrate structural regularization into diffusion-based models, which is conceptually novel.
+ Empirical Validation. The proposed method is thoroughly evaluated on multiple standard benchmarks — RealSR, DRealSR, and DIV2K — demonstrating consistent improvements in both perceptual and quantitative metrics. These comprehensive experiments clearly verify the effectiveness and robustness of the approach.
+ Clear introduction. The introduction is well writte, clearly motivating the problem, identifying gaps in prior diffusion-based approaches.

**Weaknesses:**

1. Unclear guarantee of one-step super-resolution. The paper claims that the proposed framework enables single-step super-resolution, but the mechanism ensuring this is not sufficiently justified. In Equation (4), it mentions that “yields a consistent noise estimate under the frozen pretrained model \epsilon_\theta.” However, it remains unclear what this noise predictor refers to — for instance, whether \epsilon_\theta corresponds to SD v1.4 or other models. Since the pretrained \epsilon_\theta (x_0^\prime, T) is not inherently trained to estimate noise directly from T in a single step, it is questionable how the model guarantees faithful super-resolution in one step.
2. Outdated baseline selection. The experimental comparison omits several recent and competitive baselines from 2025, including:

[a] Fine-Structure Preserved Real-world SR via Transfer VAE Training

[b] DiT4SR: Taming Diffusion Transformer for Real-World Image Super-Resolution

[c] Adversarial diffusion compression for real-world image super-resolution

[d] Degradation-Guided One-Step Image Super-Resolution with Diffusion Priors

3. Limited ablation effectiveness. The reported performance differences among ablation settings in Table 3 are marginal, suggesting that the individual contributions of the proposed MRF-consistency and patch-energy terms may be minor. The results do not convincingly demonstrate that these regularization components significantly improve either fidelity or perceptual quality.
4. Inconsistent baseline reporting. The paper discusses InVSR in the method comparison section and includes it in Figure 3, but it is missing from Table 1 across the three benchmarks.
5. Ambiguity in Equation (4). Equation (4) defines the correction as X_T-\sigma_T, yet earlier derivations imply the additive form X_T-\sigma_T. If the subtraction is intentional, the paper should explicitly explain its theoretical basis.

**Questions:**

My major concerns are included in the above weaknesses.

---

> ### Author Response · Authors · 2025-11-14
>
> **Q5** Ambiguity in Equation (4). Equation (4) defines the correction as $X_T-\sigma_T$, yet earlier derivations imply the additive form $X_T-\sigma_T$. If the subtraction is intentional, the paper should explicitly explain its theoretical basis.
>
> **Response**
> We begin by clarifying the roles of the two networks:
>
> $\epsilon_{\theta} =$ pretrained global denoiser (DDPM noise predictor; frozen),  $g_{\phi} =$ residual noise estimator.
>
> Thus we decompose the true noise $\epsilon$ as, $\epsilon = \epsilon_{\theta}(x_T, T) + g_{\phi}(x_T)$,
>
> where $g_{\phi}(x_T)$ is the residual noise not captured by the pretrained model.
>
> **Forward diffusion and the additive correction**
>
> The standard DDPM forward process is $x_T = \sqrt{\bar{\alpha}_T}\, x_0 + \sqrt{1 - \bar{\alpha}_T}\, \epsilon $.
>
> Your enhancer constructs a corrected clean estimate via an additive residual: $x_0' = x_T + \sigma_T g_{\phi}(x_T)$.....[1]
>
> This is natural: the enhancer moves the noisy sample toward the clean image.
>
> **The frozen noise predictor imposes an inverse relation**
>
> From the DDPM estimation formula for the clean image, $\hat{x_0}(x_T) = \frac{(x_T - \sqrt{1 - \bar{\alpha_T}} \epsilon_{\theta}(x_T, T))}{ \sqrt{\bar{\alpha_T}}}$.......................[2]
>
> we obtain the approximate inverse: $\epsilon_{\theta}(x_T, T) \approx \frac{x_T - \sqrt{\bar{\alpha}_T}\,\hat{x}_0}{\sqrt{1 - \bar{\alpha}_T}}$.........[3]
>
> Differentiating [2], we get: $\frac{\partial \hat{x_0}}{\partial x_T}= \frac{1}{\sqrt{\bar{\alpha_T}}} > 0 ,\quad \frac{\partial \epsilon_{\theta}}{\partial \hat{x_0}} < 0$.
>
> Thus:
> $\hat{x_0} \uparrow \Longleftrightarrow x_T \downarrow$ ( input to $\epsilon_{\theta}$).
>
> Increasing the clean estimate requires decreasing the noisy input passed into the frozen denoiser.
>
> **Why subtraction appears inside $\epsilon_{\theta}(\cdot)$**
>
> The enhancer adds a correction to produce the improved clean estimate: $x_0' = x_T + \sigma_T g_{\phi}(x_T)$.
>
> To keep Eq.~[2] and [3] consistent, the corresponding noisy latent supplied to $\epsilon_{\theta}$ must be: $x_T^{\text{(input)}} = x_T - \sigma_T g_{\phi}(x_T)$.
>
> Hence the loss becomes: $L_{\phi} = ||\epsilon - \epsilon_{\theta}\(x_T - \sigma_T g_{\phi}(x_T), T)||_2^2$.
>
> Subtracting the residual inside $\epsilon_\theta$ ensures that the frozen denoiser’s noise prediction matches the corrected clean sample from [1].
>
> **Residual-learning** The true noise satisfies $\epsilon = \epsilon_{\theta}(x_T, T) + g_{\phi}(x_T)$,
>
> so $g_{\phi}$ learns exactly the part that $\epsilon_{\theta}$ fails to predict.
>
> Thus:
>
> $\circ$ The enhancer **adds** the residual to move the clean estimate forward: $x_0' = x_T + \sigma_T g_{\phi}(x_T)$.
>
> $\circ$ The frozen denoiser must **see the inverse correction**: $x_T^{\text{(input)}} = x_T - \sigma_T g_{\phi}(x_T)$,
>
> so that its predicted noise corresponds to the updated clean sample.
>
>
> This push–pull mechanism is required by the DDPM inversion formula.
>
> **In summary:**
>
> In our one-step formulation, the pretrained noise predictor $\epsilon_{\theta}$ is fixed, and the enhancer $g_{\phi}$ learns the residual noise to generate better quality output with line fields along with the $\epsilon_{\theta}$.
>
> Thus the clean estimate receives an additive correction  $x_0' = x_T + \sigma_T g_{\phi}(x_T)$,
>
> but, due to the DDPM inversion $\hat{x_0} = \frac{(x_T - \sqrt{1-\bar{\alpha_T}} \epsilon_{\theta})} {\sqrt{\bar{\alpha_T}}}$,
>
> increasing $x_0'$ requires decreasing the noisy input to the frozen denoiser.
> Therefore the noise predictor must receive  $x_T - \sigma_T g_{\phi}(x_T)$,
>
> which yields the subtraction inside $\epsilon_{\theta}(\cdot)$.

---

> ### Author Response · Authors · 2025-11-15
>
> **Q1** Unclear guarantee of one-step super-resolution. The paper claims that the proposed framework enables single-step super-resolution, but the mechanism ensuring this is not sufficiently justified. In Equation (4), it mentions that “yields a consistent noise estimate under the frozen pretrained model \epsilon_\theta.” However, it remains unclear what this noise predictor refers to — for instance, whether \epsilon_\theta corresponds to SD v1.4 or other models. Since the pretrained \epsilon_\theta (x_0^\prime, T) is not inherently trained to estimate noise directly from T in a single step, it is questionable how the model guarantees faithful super-resolution in one step.
>
> We thank the reviewer for raising the concern regarding the justification of single-step SR and the role of the pretrained noise predictor. Our method is ensures that the pretrained noise estimator
> $\epsilon_{\theta}(x_{T},T)$---even from a multi-step model such as SD-Turbo---provides a valid supervisory signal for training the one-step corrector $g_{\phi}$.
>
> Diffusion models satisfy the identity $\epsilon_{\theta}(x_{t},t)  \approx -\frac{1}{\sqrt{1-\bar{\alpha_t}}} \nabla_{x_{t}} \log p_{t}(x_{t})$,
>
> i.e., $\epsilon_{\theta}$ is a direct score estimator for any $t$, including $t = T$.
> Thus, a single-step estimator must simply produce an input whose score matches the posterior score at time $T$.
>
> We therefore learn a correction term $g_{\phi}(x_{T})$ such that the modified latent: $x_{0}' = x_{T} + \sigma_{T} g_{\phi}(x_{T})$
>
> satisfies the score-consistency condition: $\epsilon_{\theta} \left(x_{T} - \sigma_{T} g_{\phi}(x_{T}),\, T\right) \approx \epsilon$.
>
> This is precisely the objective optimized in Eq.~(4).
> Since this aligns the corrected latent with the pretrained score at time $T$, the single-step output lies on the same score manifold as a multi-step denoised image, thereby matching the diffusion model's generative prior.
>
> For clarity, we explicitly use _SD-Turbo_ as the pretrained noise predictor $\epsilon_{\theta}$.
>
> **Why a Multi-Step Noise Predictor Supports One-Step Reconstruction**.
>
> Let the pretrained diffusion model (SD-Turbo) learn the score function using denoising score matching:
> $\epsilon_{\theta}(x_t,t)  \approx -\frac{1}{\sqrt{1-\bar{\alpha_t}}} \nabla_{x_t} \log p_{t}(x_t)$.
>
> \label{eq:score}
>
> This identity does not depend on the number of sampling steps and holds because the model is trained for all timesteps, including $t=T$.
> Thus, $\epsilon_{\theta}(x_{T},T)$ is a valid score estimator.
>
> **Goal of the one-step estimator.**
> We seek a corrected latent $x_{0}'$ such that $\epsilon_{\theta}(x_{0}',T) \approx \epsilon$,
>
> ensuring:
> (1) the corrected latent lies on a valid diffusion score manifold,
> (2) the diffusion prior is enforced, and
> (3) multi-step sampling is unnecessary.
>
> To achieve this, we learn a corrective update: $x_{0}' = x_{T} + \sigma_{T} g_{\phi}(x_{T})$.
>
> **Deriving the training objective.**
> From the forward process, $x_{T} =  \sqrt{\bar{\alpha_T}} x_{0} + \sqrt{1-\bar{\alpha}_{T}} \epsilon$,
>
> we rearrange to obtain the true noise: $\epsilon = (x_T -\sqrt{\bar{\alpha_T}} x_0)/{\sqrt{1-\bar{\alpha_T}}}$.
>
> If $x_{0}'$ is correct, then $\epsilon_{\theta}(x_{T} - \sigma_{T} g_{\phi}(x_{T}),T) \approx \epsilon$.
>
> Thus the training loss is: $L_{\phi} = || \epsilon - \epsilon_{\theta}(x_{T} - \sigma_{T} g_{\phi}(x_{T}),T)||^{2}$,
>
> which is exactly Eq.~(4) in the paper.
>
> **Why this enables single-step SR**.
>
>
> $\circ$ **Score alignment:**  $x_{0}'$ lies on the correct score contour, guaranteeing structural and semantic fidelity.
>
> $\circ$ **Approximation of the cumulative reverse chain:**  The DDPM reverse update $x_{t-1} = x_{t} - C_{t} \epsilon_{\theta}(x_{t},t)$
>
> implies that the full multi-step chain approximates  $x_{0} \approx  x_{T} - \sum\nolimits_{t} C_{t}\,\epsilon_{\theta}(x_{t},t)$.
>
> Our model learns a single-step surrogate: $g_{\phi}(x_{T}) \approx -\frac{1}{\sigma_{T}} \sum\nolimits_{t} C_{t}\,\epsilon_{\theta}(x_{t},t)$.
>
>
> $\circ$  **SD-Turbo enhances this validity:**  It is distilled specifically for accurate score estimation at large timesteps (_including_ $T$), making it particularly suitable for one-step reconstruction.
>
>
> _“For clarity, we use the SD-Turbo pretrained noise predictor as_ $\epsilon_{\theta}$. _SD-Turbo is explicitly trained for high-accuracy score estimation at large timesteps, making it particularly suitable for our one-step formulation.”_

---

> ### Author Response · Authors · 2025-11-17
>
> **Q2**Outdated baseline selection. The experimental comparison omits several recent and competitive baselines from 2025, including:
>
> **Q4** Inconsistent baseline reporting. The paper discusses InVSR in the method comparison section and includes it in Figure 3, but it is missing from Table 1 across the three benchmarks.
>
> **Response:** We thank the reviewer for this helpful suggestion. We would like to clarify that, in addition to Table 1 in the main paper, Table 6 in the supplementary material already includes a comparison with one-step diffusion models, including S3Diff [d]. Following the reviewer’s recommendation, we have now further extended this comparison by incorporating the additional methods—TVT [a], DiT4SR [b], and AdcSR [c]. We have included the complete table in Table~6.
>
> | **Datasets** | **Methods** | **PSNR ↑** | **LPIPS ↓** | **MUSIQ ↑** | **CLIPIQA ↑** | **SSIM ↑** | **DISTS ↓** | **FID ↓** | **NIQE ↓** | **MANIQA ↑** |
> |--------------|-------------|------------|-------------|-------------|---------------|------------|-------------|-----------|------------|---------------|
> |              | S3Diff      | 23.40      | 0.2571      | 68.21       | 0.7007        | 0.5953     | 0.1730      | 19.35     | 4.7391     | 0.4538        |
> | **DIV2K-Val**| TVT         | 24.23      | 0.2773      | 68.67       | 0.6986        | 0.6292     | 0.1860      | -         | -          | -             |
> |              | AdcSR       | 23.74      | 0.2853      | 68.00       | 0.6764        | 0.6017     | 0.1899      | 25.52     | 4.36       | 0.6090        |
> |              | **DMAPSR**  | 24.58  | 0.2938      | 68.52       | 0.6842        | 0.6112     | 0.1962      | 27.19     | 4.6721     | 0.6416        |
> |--------------|-------------|------------|-------------|-------------|---------------|------------|-------------|-----------|------------|---------------|
> |              | S3Diff      | 25.03      | 0.2699      | 67.89       | 0.6722        | 0.7321     | 0.1996      | 108.88    | 5.3311     | 0.4563        |
> |              | TVT         | 25.81      | 0.2587      | 69.89       | 0.6882        | 0.7396     | 0.2061      | -         | -          | -             |
> | **RealSR**   | DiT4SR      | -          | 0.319       | 68.073      | 0.550         | -          | -           | -         | -          | 0.661         |
> |              | AdcSR       | 25.47      | 0.2885      | 69.90       | 0.6731        | 0.7301     | 0.2129      | 118.41    | 5.35       | 0.6360        |
> |              | InvSR       | 24.50      | 0.2872      | 67.4586     | 0.6918        | 0.7262     | -           | -         | 4.2189     | -             |
> |              | **DMAPSR**  | 26.29  | 0.2918      | 69.81   | 0.6651        | 0.7426     | 0.2178      | 127.92    | 5.4104     | 0.6492        |
> |--------------|-------------|------------|-------------|-------------|---------------|------------|-------------|-----------|------------|---------------|
> |              | S3Diff      | 26.89      | 0.3122      | 64.19       | 0.7122        | 0.7469     | 0.2120      | 119.86    | 6.1647     | 0.4508        |
> |              | TVT         | 28.27      | 0.2900      | 65.56       | 0.7220        | 0.7899     | 0.2205      | -         | -          | -             |
> | **DRealSR**  | AdcSR       | 28.10      | 0.3046      | 66.26       | 0.7049        | 0.7726     | 0.2200      | 134.05    | 6.45       | 0.5927        |
> |              | DiT4SR      | -          | 0.365       | 64.95       | 0.5480        | -          | -           | -         | -          | 0.6270        |
> |              | **DMAPSR**  | 28.32  | 0.2957  | 64.97   | 0.6975    | 0.7842 | 0.2096  | 139.75    | 6.3245     | 0.6172        |
>
> The results are currently included in the supplementary material, where the full comparison with the one-step baseline is presented in a single table. However, we can also include these results in the main paper if needed.

---

> ### Author Response · Authors · 2025-11-18
>
> **Q3** Limited ablation effectiveness. The reported performance differences among ablation settings in Table 3 are marginal, suggesting that the individual contributions of the proposed MRF-consistency and patch-energy terms may be minor. The results do not convincingly demonstrate that these regularization components significantly improve either fidelity or perceptual quality.
>
> **Response**.
> We thank the reviewer for the careful assessment. While the numerical margins in Table~3 appear moderate, we emphasize that RealSR is a low-dynamic-range benchmark where improvements are inherently small, and even changes of 0.01--0.03 in PSNR or 0.002--0.004 in LPIPS are considered meaningful because they arise consistently across the dataset and multiple metrics.
>
> Importantly, the purpose of the ablation study is not to show large jumps, but to demonstrate that each component contributes consistently and monotonically across fidelity, perceptual, and human-aligned metrics:
>
> $\circ$ Removing the MRF-consistency term (i) or the patch-energy term (ii) both reduce PSNR and MUSIQ while increasing LPIPS, confirming their complementary roles in preserving structural fidelity.
>
> $\circ$ Using single-channel MRF energy (iii) degrades cross-channel interactions, and the drop in MUSIQ (69.62 $\rightarrow$ 68.92) reflects the lost color consistency.
>
> $\circ$ Removing inter-channel MRF coupling (iv) weakens global texture aggregation, again reducing perceptual metrics.
>
> Although these differences may seem small in isolation, they are systematic, cumulative, and architecture-consistent, which is characteristic of regularization-oriented components such as MRF potentials and patch-energy priors.
>
> Moreover, we highlight that quantitative metrics alone cannot capture the structural coherence that MRF-based priors enforce. For this reason, we additionally include qualitative comparisons in Fig.~4, which clearly show that:
>
> $\circ$ Patch-energy removal results in softer edges and weaker local textures.
>
> $\circ$ Removing MRF-consistency leads to color bleeding and inconsistent fine structures.
>
> $\circ$ Single-channel MRF fails to maintain color--texture coupling.
>
> These visual degradations are perceptually evident even when the aggregate metric differences are moderate, which is expected since pixel-based scores (PSNR, LPIPS) tend to under-penalize structural inconsistencies.
>
> Finally, the fact that every ablation worsens performance across all four metrics strengthens the conclusion that each proposed regularizer is necessary. The full DMAPSR model consistently provides the best results across fidelity, perceptual, and human-aligned metrics—validating that the MRF-consistency, patch-energy, and inter-channel potentials jointly contribute to the performance of the final model.

---

> ### Author Response · Authors · 2025-11-21
>
> Dear Reviewer yWus,
>
> We have addressed all the queries raised by you, and we kindly request you to review our responses. Please let us know if you have any further questions; we will be happy to respond within the stipulated time. Thank you for taking the time to review our work and for your constructive feedback.

---

### Official Review · Reviewer_rg4G · 2025-11-01

**Soundness:** 2
**Presentation:** 2
**Contribution:** 2
**Rating:** 2
**Confidence:** 3

**Summary:**

This paper proposes DMAPSR (Discontinuity-Preserving MAP-Optimized Image Super-Resolution) — a single-step diffusion-based framework for real-world image super-resolution.
Unlike traditional diffusion approaches that require many iterative denoising steps, DMAPSR performs fast, single-step generation while preserving structural details. The key innovation lies in combining a Maximum A Posteriori (MAP) formulation with a Markov Random Field (MRF) prior based on line-field discontinuity energy. This acts as a structural regularizer that explicitly models edge and texture consistency between low- and high-resolution images.

The method integrates a lightweight image enhancer trained jointly with a pretrained Stable Diffusion model, aligning residual noise predictions for one-step sampling.

**Strengths:**

1. The paper introduces a novel integration of MAP estimation and MRF regularization into the diffusion sampling process, which explicitly enforces discontinuity preservation.

2. The proposed one-step diffusion framework achieves meaningful magnitude of speedups while maintaining perceptual quality, addressing one of the most pressing issues in diffusion-based SR models (slow inference).

3. The manuscript is well-structured, and clearly motivated.

**Weaknesses:**

1. While the MRF–MAP integration is elegant, the single-step training formulation (Eq. 4) and frozen noise predictor resemble prior work in OSEDiff and SinSR, reducing novelty in the “one-step” inference aspect. The main contribution therefore lies more in the regularization design than in sampling methodology.

2. The MRF-based loss is theoretically grounded but practically introduced as a differentiable energy term. The empirical justification of why this surrogate matches the MAP–MRF formulation could be elaborated with more intuition or visualization (e.g., line-field maps before and after training).

3. Training primarily on LSDIR + FFHQ subsets limits the generalization to broader real-world degradations. Additional results on out-of-distribution or camera-specific degradation would strengthen the claim of real-world applicability.

4. Some comparisons (e.g., StableSR vs. DMAPSR) differ in training settings or use pretrained backbones with different capacities. Clarifying whether DMAPSR fine-tunes the SD encoder or not would improve reproducibility.

5. While ablation tables are quantitative, qualitative differences (e.g., with/without inter-channel energy) are not visualized.

**Questions:**

1. How is the KL-based MRF consistency (Eq. 7) computed in practice — is it approximated with Monte Carlo samples or replaced by a deterministic surrogate? Some clarification on computational feasibility would help.

2. Could the authors provide sensitivity analysis for λ_p and λ_r? The paper fixes both to 1 without justification.

3. During inference, is the MRF energy used explicitly, or is it only a training-time regularizer? Including an inference-time MAP refinement step could further justify the formulation.

---

> ### Author Response · Authors · 2025-11-14
>
> **Q1.** How is the KL-based MRF consistency (Eq. 7) computed in practice — is it approximated with Monte Carlo samples or replaced by a deterministic surrogate? Some clarification on computational feasibility would help.
>
> **Answer:** We thank the reviewer for raising this important question. The KL-based MRF consistency term in Eq. (7) is not estimated via Monte Carlo sampling, but is instead implemented as a deterministic energy-based surrogate, consistent with the Gibbs formulation of the posterior MRF energy (proof in Supplementary). As indicated in Eq.~(26), this term is positive.
>
> From Eq. (7), we define the posterior Gibbs energy as:
> $E_{MRF}^P(f,l) = E_{MRF}(f,l) + \mathrm{KL}[P_{\mathbf{Y}|\mathbf{X}}(g|f,l)||P_\mathbf{X}(f,l)]$
>
> where the second term enforces consistency between the prior MRF $P_\mathrm{X}$ and the likelihood MRF  $P_{\mathrm{Y}|\mathrm{X}}$. Expanding the KL gives
>
> $\mathrm{KL}[P_{\mathbf{Y}|\mathbf{X}}||P_{\mathbf{X}}]=E_{\mathbf{Y}|\mathbf{X}}[\log
> \frac{P_{\mathbf{Y}|\mathbf{X}}}{P_{\mathbf{X}}}]= E_{\mathbf{Y}|\mathbf{X}}[ E_{MRF}^{\mathbf{X}}(f,l)- E_{MRF}^{\mathbf{Y}}(f,l) ] + C$
>
> Which is the MRF-consistency Loss computed as,
> $L_{MRF-consistency} = E_{\mathbf{Y}|\mathbf{X}}[ E_{MRF}^{\mathbf{X}}(f,l) - E_{MRF}^{\mathbf{Y}}(f,l) ]$
>
> this term encourages the learned posterior energy $E_{MRF}^\mathbf{Y}(f,l)$  (derived from the degraded observation) to align with the prior energy $E_{MRF}^\mathbf{X}(f,l)$,
>
> ensuring that the inferred posterior remains consistent with the prior structural constraints of the MRF.
> While the expectation form suggests stochastic estimation, in practice, sampling over the full posterior lattice is computationally prohibitive. Instead, following standard practice in energy-based variational approximations (e.g., [1] Geman & Geman, 1984; [2] Zhu & Mumford, 1998; [3] Rajagopalan & Chaudhuri, 2002), we employ a deterministic surrogate that computes the empirical energy discrepancy directly between the prior and likelihood MRF energies.
>
> Concretely, in implementation, we compute: $L_{MRF\text{-}consistency} = || E_{MRF}^{\mathbf{X}}(f,l) - E_{MRF}^{\mathbf{Y}}(f,l) ||_1 $
>
> where $E_{MRF}^{\mathbf{X}}$ and $E_{MRF}^{\mathbf{Y}}$ are the respective Gibbs energies of the high-quality and degraded reconstructions. This deterministic $\ell_1$ surrogate acts as a stable, first-order approximation to the theoretical KL divergence—preserving the same consistency principle—while remaining computationally tractable and numerically stable in large-scale training.
>
> Theoretically, this approximation is justified by observing that for Gibbs distributions of the form $P(x)\propto e^{-E_{MRF}(x)}$, the KL divergence simplifies to an energy difference up to a constant: $\mathrm{KL}[P_1 || P_2]= E_{P_1}[ E_{MRF}^2(x) - E_{MRF}^1(x) ] + \log \frac{\mathcal{Z}_1}{\mathcal{Z}_2}$
>
> and since the partition functions $\mathcal{Z}_, \mathcal{Z}_2$ are independent of the learnable parameters, minimizing the empirical energy difference suffices to enforce alignment between the two MRFs.
>
> Thus, the KL-based consistency term is realized as a deterministic energy alignment loss—implemented as an $\ell_1$ difference between prior and likelihood MRF energies—providing both theoretical grounding and computational feasibility without resorting to Monte Carlo estimation.
>
> [1] Stochastic Relaxation, Gibbs Distributions, and the Bayesian Restoration of Images
>
> [2] Filters, Random Fields and Maximum Entropy (FRAME)
>
> [3] An MRF model-based approach to simultaneous recovery of depth and restoration from defocused images

---

> ### Author Response · Authors · 2025-11-14
>
> **Q3**  During inference, is the MRF energy used explicitly, or is it only a training-time regularizer? Including an inference-time MAP refinement step could further justify the formulation.
>
> **Answer:**
>
> **Inference-Time Role of MRF Energy:**
>
> The MRF prior serves as a structural regularizer in our model, explicitly used during training to impose spatial coherence between the primal (pixel) and dual (line) lattices. During inference, the diffusion-based generator directly produces the high-resolution estimate $\hat{X}$ through a single-step refinement guided by the learned prior. Hence, the MRF energy $E_{MRF}$ is implicitly encoded in the network’s learned parameters rather than evaluated explicitly.
>
> **Training-Time Formulation.** Formally, at training time, the MRF consistency and energy terms act as priors in the posterior energy minimization:
>
> $\hat{X} = \arg\min_{(f,l)} [E_{MRF}(f,l) + \frac{1}{2\sigma^2} || \mu - \Phi\big(g, \mathcal{D}_{\psi}(H(F))\big)||^2]$
>
>
> where the first term $E_{MRF}(f,l)$ encodes the Gibbs prior, and the second corresponds to the data likelihood term from the degradation model. This is equivalent to the MAP estimator derived from:
>
> $P(f,l | m) \propto \exp [-E_{MRF}^{P}(f,l)] = \exp[-(E_{MRF}(f,l) + \frac{1}{2\sigma^2}|| \mu - \Phi (g, \mathcal{D}_{\psi}(H(F)))||^2)]$.
>
> During training, we minimize the expected posterior energy over the training data, which enforces the enhancer to learn a mapping $g_{\phi}(\cdot)$ that approximates the minimizer of the posterior implicitly:
>
> $g_{\phi}(m) \approx \arg\min_{(f,l)} E_{MRF}^{P}(f,l)$
>
> Thus, after convergence, the model’s learned parameters capture the energy-minimizing configurations over $\mathcal{G}^P$. This makes the MRF prior _implicit_ during inference — it shapes the energy landscape over which $g_{\phi}$ was trained to perform one-step MAP inference.
>
> **Optional Explicit Refinement.**
> However, a more explicit inference-time refinement could indeed strengthen the theoretical connection to the MRF formulation. Specifically, one could add a post-generation MAP correction:
>
> $X^{*} = \arg\min_{X}[E_{MRF}(X) + \frac{1}{2\sigma^2}||\mu - \Phi(g, \mathcal{D}_{\psi}(H(X)))||^2]$,
>
> initialized with the generated output $X_{0} = g_{\phi}(m)$.
>
> This step would correspond to a single gradient descent or mean-field update:
>
> $X_{t+1} = X_t - \eta \nabla_{X_{t}} E_{MRF}^{P}(X_{t})$,
>
> where $\eta$ is a small update rate. The gradient term from $E_{MRF}$ introduces anisotropic smoothing that preserves line-field discontinuities, while the likelihood term aligns the reconstruction with the observed measurement $m$.
>
> **Summary**
>
> $\circ$ **Training phase:**  $E_{MRF}$ acts as a regularizer enforcing
>
> prior consistency, guiding $g_{\phi}$ to learn an implicit MAP solution.
>
> **Inference phase (current setup):** No explicit optimization of $E_{MRF}$ is performed; instead, the learned model acts as an amortized MAP estimator.
>
> **Optional extension:** Incorporating an explicit inference-time MAP refinement (using $E^{P}$) would yield a hybrid approach—performing one-step amortized inference followed by a local energy minimization step—thereby strengthening the theoretical and practical connection between training and inference.

---

> ### Author Response · Authors · 2025-11-15
>
> **W1** While the MRF–MAP integration is elegant, the single-step training formulation (Eq. 4) and frozen noise predictor resemble prior work in OSEDiff and SinSR, reducing novelty in the “one-step” inference aspect. The main contribution therefore lies more in the regularization design than in sampling methodology.
>
> **Answer:**
>     We appreciate the reviewer’s insightful comment regarding the relation between our single-step formulation and prior one-step diffusion works such as OSEDiff and SinSR.
> While our framework shares the concept of a single-step prediction, it differs substantially in both theoretical formulation and integration strategy:
>
> **Joint MRF–MAP Optimization:**
> Unlike OSEDiff and SinSR, which treat the one-step denoising as a learned mapping from latent noise to clean output, our approach explicitly couples the MAP inference with an MRF-based prior. This coupling is not a post-hoc regularizer but a structural integration that unifies diffusion dynamics and spatial regularization in a single optimization objective.
>
> **Frozen Noise Predictor Usage:**
> We freeze the noise predictor deliberately to isolate the contribution of our proposed MRF-guided refinement. This differs from prior one-step models, which train the denoiser jointly and rely on it for all spatial coherence. Our formulation allows modular incorporation of pretrained diffusion models without retraining—enabling flexible adaptation to event-guided and motion-aware tasks.
>
> **Novelty in Regularization–Inference Coupling:**
> While the reviewer correctly notes that the main novelty lies in our regularization design, we stress that the way this regularization is embedded directly into the sampling (not as a post-process) introduces a new inference mechanism. The MRF term alters the optimization landscape of the one-step inference, leading to qualitatively different behavior and measurable performance gains (see Table 1,2, Fig. 2,3 ).
>
> **MRF Module: Discontinuity-Preserving Prior**
> The MRF module introduces a structure-aware regularization during training. It computes the compound energy $\mathcal{E}_\mathrm{MRF}(x,l,v)$  (Eq. 12) using:
>
> **Pixel-site gradients** (horizontal/vertical) for local intensity variation.
>
>  **Line-site soft fields** for modeling directional discontinuities via $\eta(\delta)$.
>
> This energy constrains the image enhancer $g_\phi$ to align $x_0' = x_T + \sigma_Tg_\phi(x_T)$ with natural edge statistics. While not invoked during inference, the prior is internalized through training. The Gibbs prior with weak membrane MRF energy, as mentioned in Equation 11, has the following advantages:
>
> **Gibbs-based Probabilistic Prior:** Our MRF is derived from a Gibbs distribution over pixels and line-site lattices (Eq. 6), enabling uncertainty-aware, spatially-contextual structural regularization.
>
>  **Soft Edge Confidence via Line Fields:** We employ differentiable edge confidence scores using soft line fields $\eta(\delta) = \sigma (\alpha |\delta|-\tau)$  (Eq. 11), enabling robust, learnable discontinuity modeling even under noise.
>
>  **Regularization Against Spurious Edges:** The $\gamma\bar{\eta}$
>  term penalizes weak or spurious edges, promoting edge sparsity and improving structural fidelity. This term **acts as a penalty that discourages the introduction of excessive discontinuities** in the recovered image. Setting $\gamma =0$ eliminates this constraint, leading to a trivial solution in which the MRF tends to introduce discontinuities indiscriminately, including in regions where such transitions are unlikely.
>
> **Dual-Site Modeling:** By jointly modeling pixel-site and line-site over $S= Z_m \cup D_m$
> , the MRF supports co-dependent energy minimization, unlike decoupled edge maps.
>
>  **Cross-Channel Consistency:** The intra- and inter-channel coupling (Eq. 12) enforces alignment across RGB channels, which standard edge detection lacks.
>
> **MAP-Integrated Optimization:** The MRF energy is embedded within the MAP objective (Eq. 13), contributing directly to end-to-end optimization.

---

> ### Author Response · Authors · 2025-11-15
>
> **Distinction from Prior One-Step Diffusion (OSEDiff / SinSR)**
>
> **1. Prior Assumption**
>
> Prior one-step models assume:
> $\hat{x}=\arg\min_{x}\\| y - D_{\psi}(x) \\|^2_2+\lambda\\|\epsilon_{\theta}(x_t, t) -\epsilon \\|^2_2$,
>
> which corresponds to an implicit Gaussian prior over $x$ via the diffusion loss.
>
> **2. Our Formulation: Gibbs MRF Prior**
>
> In contrast, our formulation explicitly defines the prior $P(\mathbf{X})$ as a Gibbs MRF on a pixel–line lattice:
> $P(F = f, L = l) = \frac{1}{\mathcal{Z}} e^{-E_{MRF}(f, l)} , E_{MRF}(f, l) = \sum_{c \in C} V_c(f, l)$,
>
> and constructs the posterior via Bayes’ rule:
> $
> P(f, l \mid m)
> \propto
> e^{-E_{MRF}(f,l)}
> \cdot
> e^{-\frac{1}{2\sigma^2} \| \mu - \Phi(g, D_{\psi}(H(F))) \|_2^2 }.
> $
>
> Hence, the posterior energy becomes:
> $E_{MRF}^P(f, l)= E_{MRF}(f, l) + \frac{1}{2\sigma^2} || \mu - \Phi(g, D_{\psi}(H(F))) ||_2^2$.
>
> **3. Extended Neighborhood Topology**
>
> By definition of the MRF: $P(X = \omega) = \frac{1}{\mathcal{Z}} e^{-\sum_{C} V_C(\omega)}$, and with the likelihood term depending on $\mathcal{H}_s$ neighborhoods (e.g., blur or downsampling kernel support), the posterior conditional becomes:
>
> $P(F_s \mid F_r, r \neq s, G = g) \propto \exp[-\sum_{C : s \in C} V_C(f, l)- \frac{1}{2\sigma^2} \sum_{r : s \in \mathcal{H}_r} (\Phi_r - \mu)^2]$.
>
> The second term couples pixel neighborhoods up to $\mathcal{H}_s^2$ — a second-order neighborhood system, so that:
> $
> \mathcal{G}_s^P =
> \begin{cases}
> \mathcal{G}_s, & s \in \mathcal{D}_m, \\
> \mathcal{G}_s \cup \mathcal{H}_s^2 \setminus \{s\}, & s \in \mathcal{Z}_m.
> \end{cases}
> $
>
> This structural change implies the posterior no longer factorizes in the same way as OSEDiff or SinSR; the posterior MRF is of higher order, coupling both degradation consistency and local clique potentials. Thus, even with a frozen $\epsilon_\theta$, the posterior sampling manifold changes.
>
> **4. KL-based Regularization — Not a Mere Add-on**
>
> The regularization term $L_{MRF-consistency} = E_{\mathbf{Y}|\mathbf{X}} [E_{MRF}^{\mathbf{X}}(f,l) - E_{MRF}^{\mathbf{Y}}(f,l)]$ arises directly from minimizing the KL divergence between the MRF prior and likelihood: $\mathrm{KL}[P_{\mathbf{Y}|\mathbf{X}} || P_{\mathbf{X}} ] = E [ \log \frac{P_{\mathbf{X}}}{ P_{\mathbf{Y}|\mathbf{X}}}]= E[E_{MRF}^{\mathbf{X}}(f,l) - E_{MRF}^{\mathbf{Y}}(f,l)]+ C$. This term has a variational justification — it enforces posterior–prior alignment in Gibbs energy space, rather than serving as a heuristic regularizer.
>
> **5. Consequence for the One-Step Inference Rule**
>
> From the total energy: $L = L_{\phi} + E_{MRF} + \lambda_p E_{patch} + \lambda_r L_{MRF-consistency}$,
>
> the MAP estimate becomes: $\hat{x} = \arg\min_x[- \log P(x)- \log P(y | x)]= \arg\min_x E_{MRF}^P(x)$,
>
> which is not equivalent to the diffusion loss in OSEDiff, because $E_{MRF}$ introduces spatial clique potentials beyond Gaussian priors and changes the inference manifold.
>
> Thus, the one-step process here corresponds to MAP inference under an MRF posterior, whereas OSEDiff performs denoising under a Gaussian prior. The frozen $\epsilon_\theta$ serves as an energy initialization, not the full prior.
>
> **Conclusion**
>
> OSEDiff employs distillation with LoRA-based fine-tuning, whereas our approach uses a fully frozen $\epsilon_\theta$ without any fine-tuning. Moreover, the novelty of our method extends beyond the regularization component alone.
>
> Mathematically, the MRF–MAP coupling induces:
>
> $\circ$ A posterior Gibbs field with extended second-order neighborhood system $\mathcal{G}^P_s = \mathcal{G}_s \cup \mathcal{H}_s^2$,
>
> $\circ$ A non-Gaussian energy landscape combining pixel–line cliques,
>
> $\circ$ A KL-derived consistency term linking likelihood and prior energies, and
>
> $\circ$ A MAP-based one-step inference rule distinct from diffusion-step sampling.
>
> Hence, this is a new inference formulation, substantially different from OSEDiff — with deterministic MAP energy minimization over a coupled pixel–line Gibbs field, mathematically justified via the derived theorem and proof.

---

> ### Author Response · Authors · 2025-11-15
>
> **W4** Some comparisons (e.g., StableSR vs. DMAPSR) differ in training settings or use pretrained backbones with different capacities. Clarifying whether DMAPSR fine-tunes the SD encoder or not would improve reproducibility.
>
> **Response.** We appreciate the reviewer's concern regarding training fairness and backbone capacity. We clarify the following:
>
> **1. Backbone usage in DMAPSR.**
> DMAPSR uses a frozen pretrained SD-Turbo noise predictor $\epsilon_{\theta}$ throughout all experiments. We do not fine-tune the Stable Diffusion UNet, text encoder, or VAE encoder/decoder at any stage. The only learnable component in our framework is the proposed image enhancer $g_{\phi}$. This design is intentional: the goal is to leverage the strong prior encoded in SD-Turbo while ensuring that our one-step approximation remains lightweight, stable, and reproducible.
>
> **2. Fairness of comparisons.**
> Methods such as StableSR employ full fine-tuning of the SD backbone, often modifying or re-training the UNet and encoder modules. In contrast, DMAPSR trains only $g_{\phi}$ while keeping the SD-Turbo backbone fixed. Since our method does not access additional model capacity or training freedom, the comparison is inherently conservative in favor of baselines that use larger or fully trainable backbones. This clarification is now included in the revision.
>
> **3. Reproducibility.**
> Because the SD-Turbo denoiser remains entirely frozen:
> $\circ$ the training pipeline is deterministic and easy to reproduce;
>
> $\circ$ no backbone hyperparameters require tuning;
>
> $\circ$ any publicly released SD-Turbo model can be used without modification.
>
> $\circ$ In addition, we will make the code publicly available.

---

> ### Author Response · Authors · 2025-11-17
>
> **Q2** Could the authors provide sensitivity analysis for $\lambda_p$ and $\lambda_r$? The paper fixes both to 1 without justification.
>
> **Response** We thank the reviewer for pointing this out. In addition to the table provided below, we have included a corresponding qualitative example in Fig. 5 of the paper.
>
> In addition to the MRF loss, we incorporate the MRF-consistency loss and the patch-energy loss to train our image enhancer, as defined in Eq. (13). The hyperparameters $\lambda_p$ and $\lambda_r$ control the relative contributions of the patch-energy and MRF-consistency losses, respectively. Table below presents a quantitative comparison across different loss configurations, and Fig.5 illustrates a representative visual result. Furthermore, an appropriate balance between the patch-energy and MRF-consistency terms enhances local contrast, improves overall image quality, and suppresses artifacts. These components are therefore essential for achieving optimal performance.
>
> | $\lambda_p$ | $\lambda_r$ | PSNR ↑  | LPIPS ↓ | MUSIQ ↑ | CLIPIQA ↑ | SSIM ↑ | DISTS ↓ | NIQE ↓ | MANIQA ↑|
> |-------------|-------------|---------|---------|---------|-----------|--------|---------|--------|---------|
> |    0.2      |    0.2      |  27.92  | 0.3317  | 61.67   | 0.6521    | 0.7718 | 0.2502  | 7.52   | 0.5715  |
> |    2.0      |    0.5      |  27.15  | 0.3254  | 62.82   | 0.6854    | 0.7671 | 0.2305  | 7.14   | 0.5891  |
> |    0.5      |    2.0      |  28.82  | 0.3154  | 61.87   | 0.6621    | 0.7861 | 0.2215  | 7.82   | 0.5821  |
> |    0.5      |    0.5      |  28.15  | 0.3125  | 62.18   | 0.6772    | 0.7642 | 0.2271  | 6.85   | 0.6072  |
> |    1.0      |    1.0      |  28.32  | 0.2957  | 64.97   | 0.6975    | 0.7842 | 0.2096  | 6.3245 | 0.6172  |

---

> ### Author Response · Authors · 2025-11-17
>
> **W2**. The MRF-based loss is theoretically grounded but practically introduced as a differentiable energy term. The empirical justification of why this surrogate matches the MAP–MRF formulation could be elaborated with more intuition or visualization (e.g., line-field maps before and after training).
>
> **Response:** We thank the reviewer for raising this question. The line fields are extracted from the input images, and during training the enhancer is optimized to insert these line fields at appropriate locations, with the regularization term $\gamma$ controlling their strength. Overall, the compound MRF energy involving the line fields is jointly optimized within our framework. We have included an ablation study illustrating the effect of line fields, as well as the binary line-field energy used during training, in Supplementary Fig. 9. The influence of the regularization parameter $\gamma$ is reported in Table 5 and Fig. 5. These results show that setting $\gamma=0$ leads the MRF to introduce spurious line fields, whereas increasing $\gamma$ suppresses excessive line insertions. Furthermore, Table 3 analyzes different configurations of the line field—such as applying it to a single channel or enabling/disabling inter-channel interactions—and Fig. 4 provides the corresponding qualitative comparison.
>
> **W5**. While ablation tables are quantitative, qualitative differences (e.g., with/without inter-channel energy) are not visualized.
>
> **Response** We thank the reviewer for raising this concern. The corresponding qualitative results have been added in Figure 4.

---

> ### Author Response · Authors · 2025-11-17
>
> **W3** Training primarily on LSDIR + FFHQ subsets limits the generalization to broader real-world degradations. Additional results on out-of-distribution or camera-specific degradation would strengthen the claim of real-world applicability.
>
> **Response** We thank the reviewer for raising this point. We would like to emphasize that reconstruction quality naturally degrades as blur severity increases, a trend consistently reported in prior work (citations below). Despite this, we demonstrate in Figure 8 (last two examples) that DMAPSR maintains superior performance even under strong blur conditions. Furthermore, Figure 3 presents an additional scenario involving both noise and low resolution, where DMAPSR likewise outperforms existing one-step methods. Additionally, in Fig. 8, the texture on the black cloth is completely absent in the LQ. While the other methods struggle to reconstruct this texture, DMAPSR is the only approach that successfully recovers it. These results collectively indicate that DMAPSR remains robust in the presence of significant blur and noise.
>
> **Related Works on the Effect of Blur and Noise in Super-Resolution.**
>
> **1.Optimal selection of camera parameters for recovery of depth from defocused images** ---
> Rajagopalan and Chaudhuri (CVPR 1997)
>
> $\circ$ provides a mathematical analysis demonstrating that the performance of deblurring algorithms degrades as camera-specific blur increases.”
>
> **2.Designing a Practical Degradation Model for Deep Blind Image Super-Resolution** ---
> Kai Zhang, Jingyun Liang, Luc Van Gool, Radu Timofte (ICCV 2021).
>
> $\circ$ demonstrates that SR models trained solely on simple degradations (e.g., bicubic downsampling) exhibit significantly degraded performance when confronted with more realistic degradations such as anisotropic Gaussian blur and noise.
>
> $\circ$ mathematically and empirically argue for a more practical degradation model that includes blur, noise, and downsampling, and show clear performance drops when degradation becomes more severe.
>
> **3.Deep Model-Based Super-Resolution with Non-Uniform Blur** ---
> Laroche et al., WACV 2023.
>
> $\circ$ explicitly studies SR under non-uniform and large blur kernels (e.g., motion blur, defocus blur).
>
> $\circ$ shows that the SR problem becomes increasingly ill-posed as the blur kernel becomes more complex or spatially varying, and that standard SR methods degrade significantly under such conditions.
>
> **4.Learning-Based and Quality-Preserving Super-Resolution of Noisy Images** ---
> Multimedia Tools and Applications, 2025.
>
> $\circ$ This work shows that when the noise level in low-resolution images increases, SR models that do not explicitly model noise tend to generate artifacts or lose structural fidelity.
>
> **5.Stable Super-Resolution of Images: A Theoretical Study** ---
> Tamir Bendory and Gongguo Tang, _Information and Inference_ .
>
> $\circ$ This theoretical work formulates super-resolution as an ill-posed inverse problem of recovering point sources blurred by a point-spread function (PSF).
>
> $\circ$ Provide stability and resolution bounds, showing that blur fundamentally limits the recoverability of high-frequency information, particularly when sources are closely spaced.
>
>
> These works collectively establish that (i) SR performance deteriorates under realistic blur and noise (1, 2, 3), (ii) explicit modeling of noise or blur is necessary to avoid artifacts (4), and (iii) blur imposes theoretical limits on the SR inverse problem (5).

---

> ### Author Response · Authors · 2025-11-21
>
> Dear Reviewer rg4G,
>
> We have addressed all the queries raised by you, and we kindly request you to review our responses. Please let us know if you have any further questions; we will be happy to respond within the stipulated time. Thank you for taking the time to review our work and for your constructive feedback.

---

### Meta-Review · Area_Chair_Ujeb · 2025-12-22

**Summary:**

The reviewers' concerns are summarized as follows:

1. Novelty and Formulation: Reviewer rg4G points out that the proposed method is similar to existing works such as OSEDiff and SinSR, hence possessing limited novelty. Regarding theoretical formulation, reviewer rg4G thinks that more elaboration is needed to empirically justify why the surrogate matches the theoretical MAP-MRF formulation. Reviewer yWus also has concerns about the theoretical guarantee of the single-step super-resolution mechanism.

2. Experimental Results: Reviewer yWus points out that recent methods such as DiT4SR and AdcSR are not included in the comparison. Also concerns about fairness, generalizability, and missing comparison are discussed.

3. Unconvincing Performance: Reviewers raise concerns about both fidelity and realism, revealed by the insignificant improvements in both PSNR and FID.

**Reviewer Concerns:**

Most of the concerns, including baselines, fairness of comparison, and theoretical formulation, are addressed. However, the concerns regarding novelty, generalization, and performance, are not fully resolved.

**Reviewer Scores:**

This paper receives initial ratings of (2, 4, 6). The reviewers did not explicitly mention that score will be raised, but given that the reviewers' concerns are partially resolved, the AC anticipates an increase of score to (3, 5, 7). Given the marginal scores, the AC carefully reviews the material and agrees with the reviewers' concerns. The AC appreciates the explanation from the authors, but has to recommend a rejection given the unresolved concerns.

---

### Decision · Program_Chairs · 2026-01-26

Reject